



**Warming of the Willamette River, 1850–present: the effects of climate change**
**and direct human interventions**
Stefan A.Talke[1], David A.Jay[2], Heida L. Diefenderfer[3,4]
[1]Civil and Environmental Engineering, California Polytechnic State University, San Luis Obispo, California
[2]Civil and Environmental Engineering, Portland State University, Portland, Oregon
[3]Coastal Sciences Division, Pacific Northwest National Laboratory, Sequim, Washington
[4]School of Environmental and Forest Sciences, University of Washington, Seattle, Washington
*Correspondence to*: Stefan A. Talke (stalke@calpoly.edu)
**Keywords:** Water Temperature, Climate Change, River regulation, Anthropogenic Effects
**Key Points**
• A statistical model based on archival records back to the 1850s shows that average water
temperature has increased by 1.1 °C/century since the mid-19th century
• The largest increases in water temperature occur January–February (1.3 °C/century) and
the smallest in May-June (~ 0.8 °C/century)
• The number of warm water days above 20 °C has increased by ~3 weeks, matched by a
similar decrease in the number of days below 4 °C
• Approximately 30% of increased water temperature is attributable to system changes, and
70% to warming air temperature (climate change)



## Abstract

Using archival research methods, we found and combined data from multiple sources to produce a unique, 140 year record of daily water temperature ($T_w$) in the lower Willamette River, Oregon (1881– 1890, 1941– present). Additional daily weather and river flow records from the 1850s onwards are used to develop and validate a statistical regression model of $T_w$ for 1850– 2020. The model simulates the time-lagged response of $T_w$ to air temperature and river flow, and is calibrated for three distinct time periods: the late 19th, mid 20th, and early 21st centuries. Results show that $T_w$ has trended upwards at ~1.1 $^o$C /century since the mid-19th century, with the largest shift in January/February (1.3 $^o$C /century) and the smallest in May/June (~ 0.8 $^o$C /century). The duration that the river exceeds the ecologically important threshold of 20 $^o$C has increased by ~20 days since the 1800s, to ~60 d yr$^{-1}$. Moreover, cold water days below 2 $^o$C have virtually disappeared, and the river no longer freezes. Since ~1900, changes are primarily correlated with increases in air temperature ($T_w$ increase of 0.81 ±0.25 $^o$C) but also occur due to increased reservoir capacity, altered land use and river morphology, and other anthropogenic changes (0.34 ±0.12 $^o$C). Managed release of water influences $T_w$ seasonally, with an average reduction of 0.27 $^o$C and 0.56 $^o$C estimated for August and September. System changes have decreased daily variability ($\sigma$) by 0.44 $^o$C, increased thermal memory, and reduced interannual variability. These system changes fundamentally alter the response of $T_w$ to climate change, posing additional stressors on fauna.

## Short Summary

This manuscript uses archival measurements and a statistical model to show that water temperatures in the Willamette River have trended upwards since 1850, with the largest increase occurring in winter and the smallest in late spring. Approximately 30% of the increase is attributable to system changes, and 70% to warming air temperature (climate change). The number of warm water days has significantly increased, and near freezing conditions, common historically, no longer occur.

## 1.0 Introduction

Water temperatures are rising in many temperate streams and rivers, in part due to climate change (e.g., Kaushal et al., 2010). Beyond a warming climate, many additional factors influence water temperature ($T_w$), including land-use patterns, water withdrawal and return flows, reservoir storage, and other types of water-resources management (e.g., Olden & Naiman, 2010; Bottom et al., 2011). Because water temperature influences ecological processes, water quality, oxygen levels, and fish habitat and survivability (e.g., Caissie, 2006, Bottom et al., 2011), defining long-term temperature trends and understanding their causes is vital. However, with few exceptions (e.g., Webb & Noblis, 2007; Pohle et al., 2019), few $T_w$ records from the late 19th or early 20th century have been evaluated, particularly in North America (Kaushal et al., 2010). The limited availability of earlier records inhibits the ability to discern secular trends, evaluate causes, and assess impact. There is, therefore, a need to digitize and analyze archival water temperature records, such as those collected daily by the US Signal Service in the 1880s at 20+ coastal and river stations (see the Monthly Weather Review series of publications, volume 9 to 18).



In the Pacific Northwest, $T_w$ controls the long-term viability of salmon and other endangered spe-
cies (Mantua, 2010; Bottom et al., 2011, Isaak et al., 2012, Caldwell et al., 2013).  Above a
threshold of 18–21 ºC, various species of salmon, steelhead, and trout are stressed and become
more susceptible to disease (OR DEQ, 2006, Mantua, 2010).  As a result, regulations require that
the seven day average of the daily maximum temperature should not exceed 20 ºC, with a lower
threshold set for rearing and spawning streams (e.g., OR-DEQ, 2006).  An allowance of 0.3 ºC is
permitted for the sum of all anthropogenic point sources such as wastewater discharge, and non-
point sources such as loss of shading or heating in reservoirs. Hence, the Willamette River in
Portland, Oregon (Figure 1) is considered an impaired water body and out of regulatory compli-
ance for $T_w$ above 20.3 ºC (OR DEQ, 2006).
Accurately assessing and disentangling anthropogenic and climate change influences is challeng-
ing because of the large number of alterations and anthropogenic uses (e.g., diversions and dis-
charges), and feedbacks between different factors.  Compared to its natural state, the Willamette
River is more channelized, deeper, and reduced in length (particularly in upstream reaches; e.g.,
Sedell and Froggatt, 1984; Benner and Sedell, 1997; Gregory et al., 2002a).  The construction of
large storage reservoirs (Payne, 2002) has altered flow patterns and heating patterns within the
basin, and several hydroelectric projects increase $T_w$ (OR DEQ, 2006). Logging within the water-
shed reduces shading and also increases $T_w$ (Johnson & Jones, 2000).  Nonetheless, summertime
peak $T_w$ values at reservoir sites likely decreased after dam construction, because of increased
water depths; at the same time, autumn temperatures have increased (e.g., Angiletta et al., 2008;
Rounds, 2010). Below the storage reservoirs, channelization of the Willamette, deforestation of
the riparian corridor (decreased shading) (Gregory et al. 1991, Wallick et al. 2022), water diver-
sions, and storage for agriculture have also likely shifted $T_w$ (Berger et al., 2004).  Because of a
lack of in-situ data from pre-reservoir conditions, the cumulative effect of anthropogenic influ-
ence since European settlement is currently unknown (OR DEQ, 2006).
Hydrological and land-use changes in the Willamette Basin have occurred within a background
of a warming climate and hotter extremes. The summers of 2009, 2015, and 2021 were dry and
hot in the Pacific Northwest, with conditions consistent with the future climatology predicted by
climate models (e.g., Mote and Salathé, 2010, Bumbaco et al., 2013).  In 2015, snowpack was
extremely low, leading to record low streamflow in many rivers (Mote et al., 2016).  The combi-
nation of hot, dry weather and low river discharge produced elevated water temperatures, ad-
versely affecting salmon populations (Crozier et al., 2020). However, despite record heat waves
during the summer of 2021 (Portland reached a record air temperature of 46.7 ºC, about 5 ºC
above the previous all-time high), water temperatures in the Willamette River, a major tributary
of the Columbia River, did not reach the peak of 2015.
Anomalously hot years are useful for understanding processes that control $T_W$, and characteriz-
ing natural variability in the context of climate change. How anomalous were water temperatures
in coastal rivers in the Pacific Northwest during 2009, 2015 and 2021, and to what extent has cli-
mate change influenced extremes?  How much have water temperatures changed from natural,
background conditions? The dearth of long-term data complicates assessment of patterns and
trends, since weather patterns such as El Nino/La Nina and the Pacific Decadal Oscillation influ-
ence interannual and decadal variability in $T_W$ (Peterson & Kitchel, 2001). Also, the construction
of reservoirs, deforestation of the riparian corridor, irrigation diversions, and other land-use
changes are known to influence flow hydrographs and $T_w$ in other basins (e.g., Olden & Naimen,



2010). Because chronic and acute anthropogenic factors change over time, they may mask or ac-
centuate climate-induced variability and trends in degradation or recovery (NASEM 2022).
To investigate the secular changes in water temperatures caused by climate change and local an-
thropogenic influence, we construct a unique, instrument based $T_w$ data set on the lower
Willamette River (OR) that extends back to 1881, a time period with a cooler climate and unim-
peded, natural flows. Water temperature records were found and digitized from various federal,
state, and local archives, producing ~90 years of daily records stretching over a 140 year period.
Seasonal patterns and long-term trends are assessed, and their relationship to local air tempera-
tures are evaluated using a stochastic regression approach. Results show that extreme summer-
time water temperatures similar to 2009 and 2015 are found in the historical record (e.g., 1889
and 1941), and that water temperatures have frequently exceeded 20 °C during the summer, even
in the 19th century. However, on secular time scales, average water temperature is rising during
all times of the year, and the number of warm-water days is increasing. Therefore, temporal re-
fugia during the time periods most conducive to coldwater species are becoming increasingly
scarce.

## 2. Background and Methods

### 2.1 Setting

The Willamette River (Figure 1), with a mean annual discharge of 940 m³/s (1971– 2020 period),
drains approximately 29,700 square km of coastal Oregon (Figure 1; Branscomb et al., 2002). It
is the 13th largest river in the contiguous United States by volume (Wallick et al. 2022), and its
waters discharge into the larger Columbia River approximately 162km from the Pacific Ocean.
The lower Willamette River, the focus of this study (Figure 1), is an approximately 43 km long
region influenced by ocean tides during low-flow conditions and by backwater from the Colum-
bia River, particularly during spring (Helaire et al., 2019). Because of its location near the
mouth, the lower Willamette is influenced by and integrates climate changes and local anthropo-
genic changes occurring throughout the basin. Their net effect on $T_w$ is explored in this manu-
script; here, we first review the time history and magnitude of anthropogenic changes.



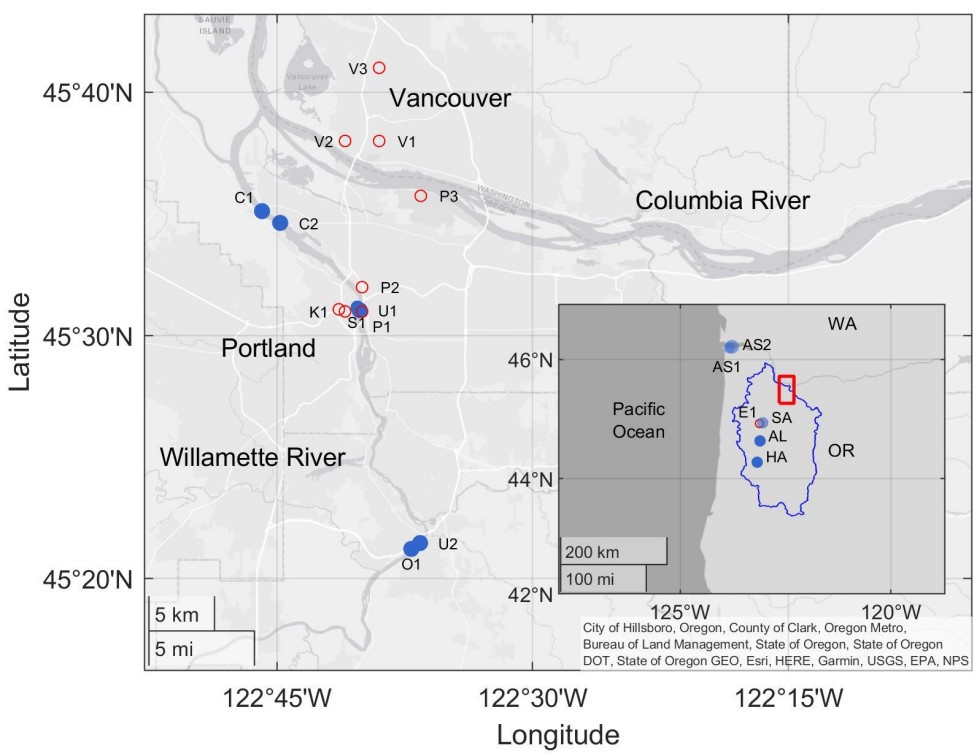


*Figure 1: Site map with locations of $T_w$ (blue, closed circles) and $T_a$ (red, opencircles) measurements. The red bounding box in the inset denotes the Portland/Vancouver Metropolitan Area depicted in the larger figure. The Willamette River watershed boundaries are denoted in blue. OR = Oregon, WA = Washington. Abbreviations and period of record of the measurements are provided in Table 1.*

The Willamette Basin has a temperate climate marked by overcast conditions from October–May, and predominately sunny, dry conditions from approximately mid-June to mid-September. Average annual precipitation on the valley floor is ~100– 130 cm/yr., with up to 500 cm occurring in the Cascade Mountains (Baker et al., 2002). Rainfall occurs primarily between October and May, with the wettest period occurring between November and January. At Portland, the largest discharge typically occurs during winter storms and peaks in the November– February period (Figure 2a). Historically, snow-melt driven flows contributed to elevated flows in the March–May time frame (Figure 2a). The combination of declining snowpack (e.g. Mote et al., 2018) and water management (e.g., Rounds, 2010) has reduced spring discharge. During summer, 60–80% of river water derives from high elevation regions above 1200m, either as direct snowmelt or as groundwater (Brooks et al., 2012). Late summer discharge has increased, however, because of the managed release of water. In the future, unimpeded wintertime discharges are expected to increase while summertime flows decrease (e.g., Chang & Jung, 2010).



The lower 300km of the Willamette River runs south-to-north through the Willamette Valley,
which is now primarily agricultural. For thousands of years the Willamette Basin was inhabited
by Native Americans, who influenced the watershed in many ways, including through controlled
burns and small-scale fish dams (Boyd, 1999, Johannessen et al. 1971, Taylor, 1999). European
settlement began in the early 1800s; Portland, founded in 1843, became the largest city in Ore-
gon by 1860 (US Census, 1866). Shading has been reduced in its modern, channelized configura-
tion compared to historical norms (Lee et al. 1995; OR-DEQ 2006). Land under irrigation was
minor before 1910, and increased from ~13,500 hectares in 1945 to about 110,000 hectares by
1979 (Sedell & Froggatt, 1984). East side tributaries such as the Clackamas River (Willamette
Rkm 40), the Mollalla River (Rkm 58) and the McKenzie River (Rkm 282) drain the mountain-
ous Cascade Range, and flow primarily through steep forested regions. West-side tributaries such
as the Tualatin River (Rkm 45) and the Long Tom River (Rkm 240) drain the lower, forested
Coast Range and are slower moving (Lee et al., 1995). The Willamette splits into the Middle
and Coast-fork at ~ Rkm 301; the headwaters of the Middle fork are approximately 486km from
the confluence of the Willamette and the Columbia rivers.
The mainstem of the Willamette River has been extensively modified since the latter part of the
19th century, first for navigation and agriculture, and later for flood control. Pre-European settle-
ment, the river was maintained in a prairie or savannah-like condition by burning (Christy and
Alverson 2011). After burning ceased (~ late 1700s), the river became fringed by a 3–7 km wide
floodplain covered by a dense riparian forest (Thilenius 1968, Sedell & Froggatt, 1984). In the
1850s, approximately 97,500ha of the Willamette Valley was mapped by the Government Land
Survey Office as riparian and wetland forest, and was dominated by tree species such as *Quercus*
*garryana* (Oregon white oak), *Fraxinus latifolia* (Oregon ash), *Acer macrophyllum* (bigleaf ma-
ple), *Alnus rubra* (red alder), and *Populus trichocarpa* (black cottonwood) (Christy and Alverson
2011). The river planform was dynamic; the upper 200km typically contained 2– 5 shallow (1.5–
3 m deep), braided channels that evolved each year due to the formation of gravel bars and drift-
wood barriers (Sedell & Froggatt, 1984; Gregory et al., 2002a; Wallick et al. 2022). Beginning in
the 1870s, but particularly in the first half of the 20th century, the river was reduced to a primar-
ily single-thread stream, and shortened by nearly 20km (Sedell & Froggatt, 1984; Gregory et al.,
2002a). Bank-stabilization measures began in the late 1800s and occurred most prominently dur-
ing the mid-20th century (1930s– 1960s); approximately 25% of Willamette River banks now
have revetments, armoring, wing dikes, and other bank protection measures (Gregory et al.,
2002b). Further, from 1870– 1950, approximately 65,000 "snags" (30– 60m long trees with a di-
ameter of 0.5– 2m) were removed (>500 per km; Sedell & Froggatt, 1984). Peak snag removal
occurred in the late 1800s/early 1900s (Sedell & Froggatt, 1984). These snags were often used to
block-up side channels. As a result, off-channel areas such as alcoves and sloughs—often 2– 7
℃ cooler than the mainstem—have decreased in extent by 70– 80% (Landers et al., 2002). Ad-
ditionally, the forested area in the floodplain has decreased by 75-90% (Landers et al., 2002,
Gregory et al. 2019). Dredging further altered the river, after its authorization in 1906. Between
1908– 1929, approximately 78,000 m³ yr⁻¹ of sediment were removed from the river above tide-
water (Willingham, 1983), but much more extensive dredging has occurred in Portland Harbor.
The depth of the river is currently ~ 12m in the lower ~20km of the Willamette, the focus area of
our study (Figure 1). Depths gradually reduce to a centerline depth as shallow as 1.5– 2m
around Rkm 280 (US Geological Survey (USGS), 2003).



A total of 371 reservoirs and impoundments of various size have been built in the Willamette ba-
sin, with a combined capacity of more than 3.3 km$^3$ (Payne, 2002). Given a mean discharge of
about 980 m$^3$s$^{-1}$ (Naik and Jay, 2011), these reservoirs store ~10.6% of the annual average flow.
The majority were built between 1950– 1980, with ~23 built pre-1950 and ~25 after 1980
(Payne, 2002). Approximately 45% are small storage reservoirs for irrigation (order 100,000 m$^3$
capacity); hydroelectric dams (~9%) and water supply reservoirs (6% of total) are typically of
similar size (Payne, 2002). A total of 13 federal reservoirs for storage and flood control were
built between 1941 and 1969 with a combined maximum storage capacity of 2.75 km$^3$ (Rounds,
2010); the largest are Detroit Dam (completed 1953, capacity 0.56 km$^3$), Lookout Point Dam
(completed 1954, capacity 0.59 km$^3$) and Green Peter Dam (0.53 km$^3$ capacity, completed 1968;
Payne, 2002; Rounds, 2010). The two federal reservoirs built in the 1940s were relatively small
(combined capacity of 0.18 km$^3$) compared to modern capacity; therefore, we consider the period
before 1953 to be pre-river flow regulation. An examination of hydrological records suggests
that flood control exerted some influence in the 1954– 1964 period, reducing peak flows during
the December 1964 flood considerably, and that the modern hydrological regime began ~1965–
1970 (Gregory et al., 2002c). In total, reservoirs have increased the surface area of water within
the system by about 200 km$^2$, with the majority (80– 85%) occurring in the 13 federally operated
water projects (Payne, 2002). A net increase of ~50 km$^2$ in water surface area is estimated for the
Willamette Valley since 1851 (Gregory et al., 2002d), in part from water impoundments. By
comparison, channelization between 1850 and 1995 only removed ~ 17 km$^2$ of water surface on
the mainstem Willamette, from 76 to 59 km$^2$ (Gregory, 2002a). Combined with the loss of ripar-
ian corridor shading during the growing season (Gregory et al., 2002e; Rounds, 2007), the in-
creased surface area in the basin means that heat input into the fluvial system—for the same me-
teorological conditions—has increased.

## 2.2 In-situ water temperature measurements

A number of measurements were obtained to assess changes to meteorological and fluvial condi-
tions since the mid-19$^{th}$ century (Figure 1; Table 1 & Table 2), and approximately 30 years of ar-
chival records were digitized. From 1881– 1890, the US Signal Service (USSS) measured top-
and bottom $T_w$ at Portland at 11:00 (local time) every day. The successor to the USSS, the US
Weather Bureau (USWB) measured $T_w$ from 1941– 1961 between 6:30 am and 7:30 am daily
(local standard time). We digitized and quality assured the previously unanalyzed USSS and
USWB records, which were obtained from the National Centers for Environmental Information
(NCEI). A spot-check of US Army Corps of Engineers records from Willamette Rkm 10.5 from
1941– 42 (Moore, 1968) showed a general consistency with USWB measurements, to within 1$^o$
C. Measurements of $T_w$ are available from the US Geological Survey (USGS) since 1961, with
~26 station years available in the Portland metropolitan area since 1971 (Table 1). Such federal
records are supplemented by additional state and local records. Intermittent Grab-sample meas-
urements of $T_w$ are available from the State of Oregon Department of Water Quality, particularly
during summer (1949, 1953– present; obtained from the City of Portland). Nearly continuous
daily measurements of $T_w$ at the Willamette Falls fish ladder from 1985– 2020 were obtained
from the Oregon Department of Fish and Wildlife. Finally, a long, continuous record has been
made available by the City of Portland at half-hourly increments from 1992– 1999 and 1997–





2015 at the Saint Johns Bridge and the St Johns Railroad Bridge, respectively (see also Annear et
al., 2003).
Water temperature records from these different locations are combined together to obtain a 90
year record of in-situ $T_w$ covering 64% of the 1881 to 2021 period (Table 1). Once-a-day meas-
urements were adjusted to the daily minimum temperature, because most historical measure-
ments were made in the morning. The adjustment, typically ~0.1 ᵒC, was based on the monthly
averaged differences between measurement time stamps and the daily minimum in modern, high
resolution data (Table 1). The composite 1881– 2021record uses lower Willamette records when
available, and the nearest data otherwise (if available). Records in Oregon City and further up-
stream were adjusted for spatial heating effects through the use of monthly averaged gradients
observed between coterminous measurements from 2000– 2017.  Most adjustments for spatial
variability were minor (<0.3 ᵒC), except for a few years (1962, 1983– 1984) in which the only
available measurements were from the middle or upper Willamette River.  Additional notes are
included in Table 1, and the source of data in the composite are included in the data record (see
supplement).
Additionally, we use $T_w$ measurements from the lower Columbia River to check our model esti-
mates (see section 2.4) during periods with no other data (Figure 1, Table 1). Water temperature
was measured up to twice daily at Astoria from 1854– 1876 (Talke et al., 2020), approximately
24 km from the present-day mouth. Monthly estimates of $T_w$ at Astoria, Tongue Point (Rkm 29)
are available from 1925– 1964 (USC&GS, 1967), and daily records were obtained from 1940–
42 (Moore, 1968) and 1949– present from the National Oceanographic and Atmospheric Admin-
istration. Before 1950, surface waters at Astoria were generally freshwater or brackish during
typical flow conditions (Al-Bahadily, 2020, USC&GS, 1967), and therefore approximate river
water temperatures.  During the November– April rainy season, good agreement is found be-
tween model results and Astoria measurements, thus helping to validate the model.  During other
times of year, snow melt from the interior Columbia River basin dominates the river flow signal
(e.g., Naik & Jay, 2011; Helaire et al., 2019), suppressing water temperature (see Results, Sec-
tion 3).  Additional information about the Astoria measurements is given in Talke et al. (2020)
and Scott et al. (2022).
Monthly averages of the USGS, DEQ, and City of Portland data from 2009 to 2015 agree to
within 0.1– 0.2 ᵒC, indicating that modern measurements from the last two decades are con-
sistent and of high quality.  This comparison also shows that grab samples from the water surface
compare favorably with other methods.  Measurements by the USSS (1881– 1890) and USWB
(1941– 1961) were made at a 1ˢᵗ-order weather station by trained professionals, and appear to be
of high quality; however, little independent verification is possible.  Evaluation of data from
1962 to the mid-1990s indicates some periods with lesser quality in which different measure-
ments disagree with each other. For example, summertime measurements from a thermograph in
Oregon City (1963– 1967) are as much as 1.8 ᵒC higher (monthly average) with coterminous
grab-samples; a smaller, but still significant, bias is found between Saint Johns Bridge measure-
ments (1971– 1975) and grab-samples (Table 1). Since the typical difference between such
measurements is reported to be <1 ᵒF (0.56 ᵒC) (Moore, 1967), some unknown issue occurred.
The availability and quality of in-situ data informs our choice of model calibration periods and
interpretation of model/data comparisons.



*Table 1: In-situ water temperature measurements used to obtain a composite record of daily minimum water tem-*
*perature in Portland, 1881– 2020. Locations ordered based on start-date and originating agency.*

| Location | Originating agency | Short name | River km | Latitude | Longitude | Measurement Dates | Measurement Frequency | Precision | Bias Correction |
|---|---|---|---|---|---|---|---|---|---|
| **Astoria Downtown**[a] | US Coast Survey | A1 | CR. 24 | 46.19 | -123.829 | 6/1854– 10/1876 | Various, usually 6:00 am and 6:00 pm daily | ±0.03 °C | None applied |
| **Stark Street, Portland**[b] | US Signal Service | S1 | 21 | 45.519 | -122.671 | 9/1881 – 11/1890 | 11:00 am daily | ±0.3 °C | 0.1 °C to 0.2 °C |
| **Astoria Tongue Point** | US CGS (pre-1973) & NOAA | A2 | CR 29 | 46.207 | -123.768 | 1/1925– present; daily to 1995, hourly 1995– present | Monthly 1/1925– 12/1964; Daily 11/1940– 6/1942, 01/1949– 12/1995; Hourly 11/1993– present | ±0.2 °C before 1994; ±0.03 °C modern | None applied |
| **Morrison Street Bridge, Portland**[b] | US Weather Bureau | W1 | 21 | 45.517 | -122.668 | 7/1941 – 10/1961 | 7:30 am daily (except Sunday) | ±0.3 °C | 0 °C to 0.2 °C |
| **Lower Willamette River**[d] | Oregon Department of Environmental Quality | D1 | 19– 21 (primarily) | Various | Various | 1949– 2015; 2746 grab samples retained after quality assurance | 6:00am– 12:00 pm; mode = 9:00 am. monthly in winter, once weekly in summertime | ±0.1 °C | Median 0.1 °C; 90% corrections < 0.2 °C |
| **Harrisburg** | USGS Gauge 14166000 | | 259 | 44.2704 | -123.174 | 6/1961– 9/1987 10/2000– Present | Daily Max, Min & Mean | ±0.05 °C | |
| **Oregon City** | USGS Gauge 14207770 | U2 | 42 | 45.3578 | -122.610 | 3/1963– 9/1967 | Daily Max, Min & Mean | ±0.05 °C | 0.7– 1.8°C Diff. w/Grab samples during summer |
| **Salem** | USGS Gauge 14191000 | SA | 137 | 44.9442 | 123.0429 | 10/1963 – 9/1987 | Daily Max, Min & Mean | ±0.05 °C | |
| **Saint Johns Bridge** | USGS Gauge 14211805 | U3 | 9 | 45.583 | -122.759 | 10/1971– 9/1975 | Daily Max, Min & Mean | ±0.05 °C | 0.6– 1.05 °C Diff. w/Grab samples during summer |
| **Morrison Street Bridge, Portland** | USGS Gauge 14211720 | U1 | 21 | 45.5175 | -122.669 | 11/1975– 9/1981 11/2001– 9/2005 01/2009– Present | Daily Max, Min & Mean through 2005. Every 30 minutes | ±0.05 °C | None applied |
| **Willamette Falls Fish Ladder**[e] | Oregon Department of Fish and Game | O1 | 43 | 45.354 | -122.618 | 01/1985– present | Not tabulated; Daily, with gaps | ± 0.2 °C | -0.3 to 0.3 °C, based on monthly difference with Portland |
| **Saint Johns Bridge**[f] | City of Portland, BES | C1 | 9 | 45.585 | -122.765 | 7/1992 – 9/1999 | Every 30 minutes | ± 0.01 °C | Very biased; not used. |
| **Saint Johns Railroad Bridge**[f] | City of Portland, BES | C2 | 11 | 45.5773 | -122.747 | 9/1997– 9/2012 | Every 15 minutes | ± 0.01 °C | Averaged with USGS record |
| **Albany** | USGS Gauge 14174000 | AL | 192 | 44.6388 | -123.107 | 08/2001– Present | Daily Max, Min & Mean | ±0.05 °C | |

Notes:  Stations ordered by start date, with earliest measurements first. All times given in local standard time. Bias corrections are subtracted
from raw measurements on a monthly basis to obtain daily minimum; a positive value indicates a downward adjustment. Coordinates provided in
the North American Datum of 1983. The locations for the measurements at Stark Street, Astoria Downtown, Willamette Fish ladder and the City
of Portland measurements are estimated based on available data. River km are the thalweg distance from the mouth of the Willamette, except for
Astoria which is on the Columbia River.
*Specific Footnotes:* (a) Measurements obtained from US National Archives; see Talke et al., 2020; (b) Measurements obtained from National
Centers for Environmental Information; (c) Data obtained from NOAA; Grab samples from 1925– 1995, approximately daily, generally between
10:00am –  1:00pm; median ~11:30 am.(d)  Data obtained from US EPA Storet database. Measurements often made from bridges in the Portland
Metro area, including the Hawthorne Bridge, the Steel Bridge, and SPSS Railway Bridge. Samples pre-1960 discarded because of lack of time
stamp.  Grab samples after 12:00 pm (noon) not considered to avoid afternoon heating signal. Pre-12:00 pm data adjusted to daily minimum on
monthly basis based on modern USGS data.  Measurements at 1– 3 day frequency in 1964– 1972; (e) Data from 1985– 1999 obtained directly
from agency; post 1999 records available online. Based on a comparison using 2001-2004 data, an average warming of 0.2 to 0.3 °C occurs be-
tween Willamette Falls and Portland from July to September.  A cooling of up to 0.3 °C occurs between March to May. Little variation occurs at
other times; (f) Obtained directly from agency; pre-2000 data also obtained from Berger et al., 2004.





### 2.2.2 Meteorological and Flow records

A nearly complete record of discharge in the lower Willamette River is available from 1893–
present, with less certain estimates from 1853– 1892. Daily discharge is available from the
USGS in Portland from 1972 to the present (USGS Gauge 14211720). Routed estimates of dis-
charge are available for earlier periods from 1878 forward from Jay & Naik (2011), based on
USGS measurements at Albany (USGS Gauge 14174000) and Salem (USGS gauge 14191000).
Routed estimates pre-1893 are less certain, because of gaps in the record (Jay & Naik, 2011).
Daily Portland water level measurements are available from 1876– present, and estimates of 30d
averaged Portland water level are available from 1855– 1876 based on tidal measurements at As-
toria (Talke et al. 2020). Nineteenth century measurements incorporate a substantial backwater
effect from the Columbia River that historically varied from zero to as much as 10 m during
some spring freshet events (see Helaire et al., 2019).

Records of daily maximum $T_a$ from the Portland-Vancouver area were found in several sources
(Table 2). Continuous daily weather records at Vancouver (1849– 1868) and Eola (1870– 1892)
were measured by the USSS and were provided in digital form by the Midwestern Regional Cli-
mate Center (https://mrcc.purdue.edu/ ). Additional daily records from the USWB and the Na-
tional Weather Service from Portland and Vancouver cover the 1874– present period and were
obtained from NCEI.

Air temperature records were carefully evaluated for potential bias (e.g., caused by elevation dif-
ferences) and consistency with each other (Table 2; see Figure 1 for locations). For example, the
Vancouver record from 1895– 1965 is on average ~0.4 to 0.5 °C warmer than the downtown
Portland record. The Portland Airport reading was <0.05 °C cooler than the downtown Portland
Weather Bureau reading between 1940– 1948, on average. Thereafter, the Portland Weather Bu-
reau record warmed more quickly, and was 0.54 °C warmer than the Airport from 1960– 1969.
The modern Portland KGW record (1973– present), located at 48.5m above sea-level, is slightly
cooler from 1991– 2020 (annually averaged daily maximum = 17.08 °C) than the Portland Air-
port (17.47 °C). Under standard atmospheric conditions, with a lapse rate 6.5 °C per 1000m, a
difference of ~0.3 °C is expected between these records. Thus, we conclude that the measured
difference between the stations is almost entirely explainable by elevation effects. After adjust-
ing for mean biases, the root-mean-square error observed between the different Portland air tem-
perature records is around 1– 1.1 °C from 1940– present. Daily maxima between Vancouver and
Portland show more variability (RMSE of ~1.5– 1.6 °C), possibly because of small differences in
climate. The influence of these small differences on our $T_w$ model results are explored later.



*Table 2: Meteorological stations used to develop statistical models, and associated root mean square*
*error (RMSE) of water temperature obtained for different calibration periods (annual, summer, and win-*
*ter). The RMSE represents either the daily or monthly averaged difference with in-situ water temperature*
*measurements, in degrees Celsius. Station Identification numbers (ID) are from the US National Weather*
*Service. Measurement dates denote the time period that daily maximum temperature was recorded at*
*the given location. The latitude/longitude value for Eola (near Salem, Oregon) is estimated. All stations*
*except Vancouver are in Oregon.*

| Name | Station ID | Measurement Dates | latitude | longitude | Model Name | Calibration Period | RMSE Annual Calibration (ºC) | RMSE Summer Calibration (ºC) | RMSE Winter Calibration (ºC) | RMSE Annual (monthly avg) (ºC) | RMSE Summer (monthly avg) (ºC) | RMSE Winter (monthly avg) (ºC) |
|---|---|---|---|---|---|---|---|---|---|---|---|---|
| Portland Downtown | USW00024274 | 1874– 1902 | 45.5166 | -122.6667 | 1881D | 1881– 1890 | 1.1 | 1.2 | 0.87 | 0.78 | 0.92 | 0.5 |
| Portland Downtown | USW00024274 | 1902– 1973 | 45.5333 | -122.6667 | 1941D | 1941– 1952 | 0.91 | 0.68 | 0.75 | 0.62 | 0.48 | 0.43 |
| Portland Airport | USW00024229 | 1938– 2021 | 45.5958 | -122.6093 | 1941A | 1941– 1952 | 0.91 | 0.66 | 0.78 | 0.6 | 0.46 | 0.42 |
| Portland Airport | USW00024229 | 1938– 2021 | 45.5958 | -122.6093 | 2000A | 2000– 2015 | 0.88 | 0.51 | 0.75 | 0.62 | 0.31 | 0.48 |
| Portland KGW[2] | USC00356749 | 1973– 2021 | 45.5181 | -122.6894 | 2000D | 2000– 2015 | 0.87 | 0.53 | 0.72 | 0.62 | 0.33 | 0.46 |
| Vancouver, Washington[3] | USC00458773 | 1849– 1868 1891– 1966 | 45.6333 | -122.6833 | 1941V | 1941– 1952 | 0.98 | 0.75 | 0.85 | 0.68 | 0.54 | 0.48 |
| Eola | US Signal Service Observation | 1870– 1892 | 44.9323 | -123.1198 | 1881E | 1881– 1890 | 1.22 | 1.41 | 1.05 | 0.91 | 1.17 | 0.72 |

Notes:

1. The annual RMSE between measurements and the climatological average is 1.86, 1.46, and 1.43 ºC for the 1881– 1890, 1941– 1952, and 2000– 2015 calibration periods, respectively.

2. The 1973– 1999 measurement was at a slightly different location of (45.517W, -122.683E). The elevation of the 1973– present dataset is ~48.5m. The lapse rate for the standard atmosphere (6.5 ºC per 1000m) suggests that the difference to a measurement at sea-level is ~0.3 ºC. An observed difference in average daily maximum temperature at the Portland Airport (17.46 ºC, <10m relative to sea-level) and Portland KGW (17.07 ºC) between 2000– 2020 is therefore mostly caused by elevation differences.

3. The Dec. 1849– 1868 measurement at Fort Vancouver was made by the US Signal Service; the approximate location was 45.633N,- 122.65E, and was several km east of the 1891– 1966 measurement. The gauge was moved in 1966 to a higher elevation location with a known bias (Mote et al., 2002). The 1966– present data is therefore not used.

## 2.3 Advection-Diffusion equation

To develop our statistical model approach, understand its limitations, and motivate its form, we
first consider the underlying physical dynamics. Heating and cooling of river water is governed
by the Advection-Diffusion equation (ADE; e.g., Fischer et al., 1979). When vertical and cross-
sectional variations in $T_w$ are neglected, the 1-D ADE for $T_w$ as a function of time $t$ and along-
channel coordinate $x$ (positive downstream) reads:



$$\underbrace{\frac{\partial T_w}{\partial t}}_{} = \underbrace{-u\frac{\partial T_w}{\partial x}}_{Advective\ Term} + \underbrace{\frac{\partial}{\partial x}\left(K\frac{\partial T_w}{\partial x}\right)}_{Diffusive\ Term} + \underbrace{\frac{H}{\rho c_p d}}_{Heating\ term}, \tag{1}$$

where $K$ is a horizontal diffusion coefficient, $u$ is river velocity, $H$ is the sum of heat flux into or out of the system, $d$ is the cross-sectionally averaged depth, and $c_p$ is the heat capacity of water, and is approximately constant to within 1% for typical variations in $T_w$. This simple ADE does not consider groundwater flow, which cools the off-channel alcoves of the Willamette River during summer (Faulkner et al., 2020).

Scaling provides insight into the relative importance of the advection, diffusion, and heating terms, relative to the time rate of change $\frac{\partial T_w}{\partial t}$. Over a 12 hour time scale during the day, temperatures in summer are observed to vary by ~0.5 °C, yielding $\left(\frac{\partial T_w}{\partial t}\right)_{daily}$ ~10⁻⁵ °C/s. Over a month, larger changes of order 5 °C are observed, yielding $\left(\frac{\partial T_w}{\partial t}\right)_{monthly}$ ~2×10⁻⁶ °C/s. The time rate of change for daily and monthly time scales must be balanced by the terms on the right hand side of Equation (1). An evaluation of measurements suggests that:

- The diffusive term is negligible. Over most of the year, the monthly average of daily $\frac{\partial T_w}{\partial x}$ is << 10⁻⁵ °C/m, except from July– September when a monthly-averaged increase of 1– 2 °C per 100km is observed (Figure 2b). Using 100km as a typical length scaleand $K$ ~1000 m²/s for the diffusive term, the $\frac{\partial}{\partial x}\left(K\frac{\partial T_w}{\partial x}\right)$ term is generally < 10⁻⁷ °C/s, much less than $\frac{\partial T_w}{\partial t}$.

- The nonlinear advective term is likely influential during summer, due to a positive $\frac{\partial T_w}{\partial x}$ (Figure 2b). During other seasons, river discharge can either cool or warm Portland water because of the presence of both negative and positive $\frac{\partial T_w}{\partial x}$ (Figure 2). Therefore, the net influence of the advective term on monthly averaged temperatures is likely small, though it may matter during weather events (such as a rain-on-snow event).

- Seasonal variations in discharge (Figure 2a) influence the magnitude of the advective term. During early summertime (June) conditions, Lee (1995) measured velocities of ~0.8 m/s in the upper Willamette; tidally averaged currents are typically 0.05– 0.1 m/s during the same period in Portland (USGS Gauge 14211720). Since discharge is smallest during August/September, the decrease in $u$ counteracts the increase in $\frac{\partial T_w}{\partial x}$ in the advective term $u\frac{\partial T_w}{\partial x}$. Overall, considering typical magnitudes of $u$ and $\frac{\partial T_w}{\partial x}$, we find that the advective term scales as 10⁻⁵ °C/s to 10⁻⁶ °C/s during the summer, depending on location.

- Based on the considerations above, the heating term is usually the leading order term that drives the time rate of $T_w$, as also found, for example, by Wagner et al., (2011).

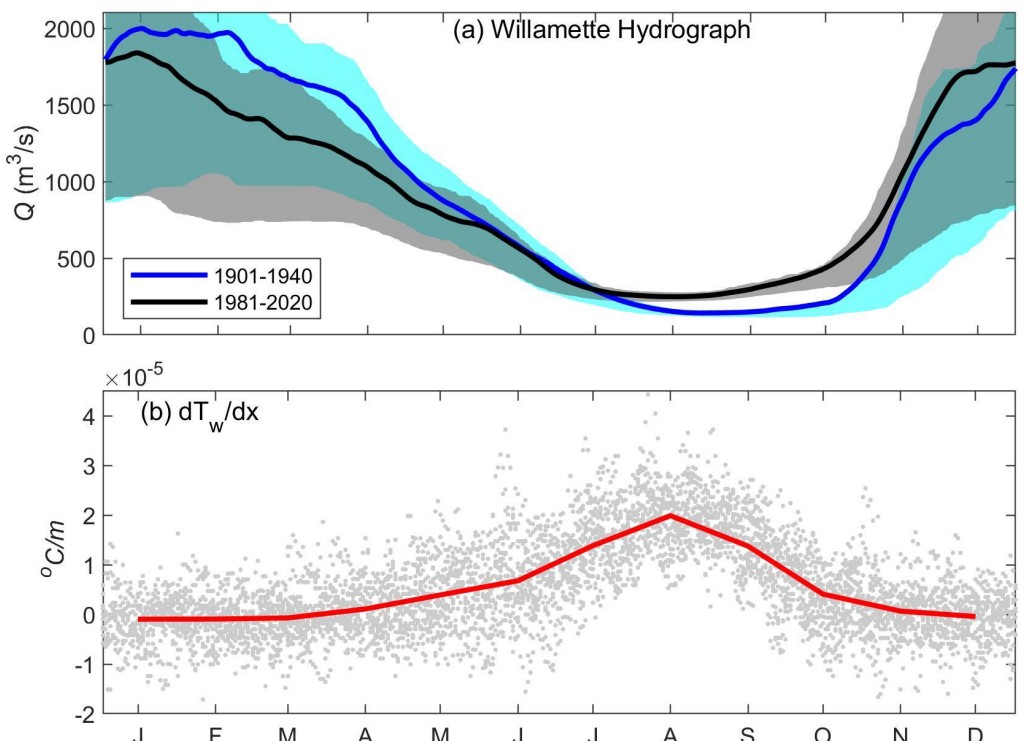

389

*Figure 2: (a) The Willamette hydrograph at Portland, Oregon for the pre-reservoir (1901– 1940) and*
*modern (1981– 2020) periods, and (b) the horizontal $T_w$ gradient between Albany, Oregon and Portland*
*Oregon for the 2000– 2017 time period. Positive indicates that downstream measurements in Portland*
*are warmer. Shading in (a) denotes the 25th and 75th percentile of measured discharge. The along-river*
*distance between Portland and Albany is 169 km. The red line in (b) denotes the monthly average. Tick*
*marks denote the middle of each month.*

When advection and diffusion are unimportant, the non-linear heating term ($\frac{H}{\rho c_p d}$) governs the
time rate of change of temperature, $\frac{\partial T_w}{\partial t}$. The $\frac{H}{\rho c_p d}$ term can be linearized using a number of as-
sumptions, enabling use of a linear regression approach in which $T_w$ is a function of $T_a$ and river
discharge $Q$. The details, described briefly below, reveal some inherent limitations. See
Mohseni & Stefan (1999) for a more detailed discussion of linearization assumptions.


First, we make the approximation that the reciprocal of depth, 1/d, is a function of $Q$:

$$\frac{1}{d} \approx a_1 - a_2 Q, \tag{2}$$



where $a_1$ and $a_2$ are constants. The negative sign reflects the observation that 1/d decreases
(depth increases) as discharge $Q$ increases.
Further, the heat flux term is a function of at least 5 different terms (e.g., Fischer et al., 1979):
$$\sum H = H_s + H_e + H_{LW,gain} + H_{LW,loss} + H_{sw} .$$  (3)
The sensible heat flux is proportional to the difference between air temperature $T_a$ and $T_w$ (both
measured in Celsius):
$$H_s = k_1 w (T_a - T_w),$$  (4)
where $k_1$ is a constant that depends on air density and several empirical coefficients, and w is the
wind speed at 10m.  The energy loss because of evaporative heat flux, $H_e$, depends on wind
speed, the latent heat of evaporation, and atmospheric conditions, and is generally small in win-
ter but potentially significant in summer (Wagner et al., 2011). The third term, the heat input
from radiation from water vapor, is
$$H_{LW,gain} = k_{LW,gain}(273.15 + T_a)^6 \propto k_{LW,gain} T_a ,$$  (5)
Where $k_{LW,gain}$ is a constant that depends on cloud cover. When $\Delta T_a$ is small relative to
$(273.15 + T_a)$, such as occurs in the Willamette, Equation 5 is approximately linear with respect
to $T_a$ . Similarly, heat loss due to long-wave radiation is modeled as
$$H_{LW,loss} = k_{LW,loss}(273.15 + T_a)^4 \propto k_{LW,loss} T_a,$$  (6)
where the power term is approximately linear in $T_a$ for temperature differences < 20 degrees Cel-
sius (see also Mohseni & Stefan, 1999).  Finally, the heat input from incoming shortwave radia-
tion, $H_{sw}$, is a function of sun angle, albedo, and atmospheric effects. Wagner et al. (2011) used
the climatologically averaged insolation as a basis function in their $T_w$ model, but most models
implicitly assume that $H_{sw} R$ is proportional to $T_a$, (Benyahya et al., 2007).
Combining Equations 3 to 6, and neglecting the evaporation term, we find that $H$ can be linear-
ized as follows:
$$H(t) \approx b_1 T_a + b_2 T_w + b_3 + error,$$  (7)
where $b_1$, $b_2$, and $b_3$ are constants.
Combining Equation 7 and Equation 2, the heating term can be approximated by:
$$\frac{H}{\rho c_p d} \approx c_1 T_a + c_2 T_w - c_3 Q T_w + c_4 Q T_a + \epsilon,$$  (8)
Where $\epsilon$  is the approximation error and $c_1$, $c_2$, $c_3$, and $c_4$ are coefficients.  Equation 8 shows that
even after many simplifications and approximations, there are still nonlinear interactions be-
tween terms such as air temperature and river flow (i.e., the $Q T_a$ term).   In practice, it is found



or assumed that air temperature is the most important factor in heating, and only the $T_a$ depend-
ence is retained (e.g., Erickson & Stefan, 2000, Webb et al., 2003). Most statistical models im-
plicitly start with this assumption, though some non-linear regression approaches have been ap-
plied (see review by Benyahya et al., 2007). For our purposes here, we note that simplifying
heating to be a linear function of $T_a$ works best during periods of relatively constant water tem-
peratures and river discharge (see also Mohseni & Stefan, 1999). This is one reason why models
calibrated to a specific season such as summer often works better than a model fit to an entire
year (see below).
The advection term in Equation 1 can similarly be linearized by assuming that either $\frac{\partial T_w}{\partial x}$ or $Q$ is
constant or slowly varying, relative to the other. This yields either a regression term in $Q$ or in
$T_w$. Removing nonlinear terms, the following linearized basis function emerges:
$$\frac{\partial T_w}{\partial t} = b_w T_w + b_a T_a - c_Q Q, \qquad (9)$$
where $b_w$, $b_a$, and $c_Q$ are coefficients and the minus sign indicates that river flow reduces water
temperature. Using the approximation $\frac{\partial T_w}{\partial t} \approx \frac{T_{wn} - T_{wn-1}}{\Delta t}$, we find that $T_w$ at time step $n$ is equal to
the $T_w$ at the previous time step $n-1$, plus a correction that is a function of $T_a$ and $Q$:
$$T_{w_n} = T_{w_{n-1}} + \Delta t b_w T_{w_n} + \Delta t b_a T_a - \Delta t c_Q Q \qquad (10)$$
This is an autoregressive (AR1) process. Hence, at time $n-1$, $T_w$ is a function of the $T_w$ at time
$n-2$, and the $T_w$ at $n-2$ depends on $T_w$ at $n-3$. If we develop and then substitute the solutions
for $T_{w_{n-1}}, T_{w_{n-2}}, \ldots$ into Equation 10, we find that
$$T_w(t) = \sum_{\tau=0}^{\tau=j} a_\tau(t-\tau) T_a(t-\tau) + \sum_{\tau=0}^{\tau=j} b_\tau(t-\tau) Q(t-\tau) + C, \qquad (11)$$
where $a_\tau$ and $b_\tau$ are regression coefficients at some time lag $\tau$, $C$ is a constant of regression,
and the time period $j$ is chosen to be long enough that the coefficients $a_\tau$ and $b_\tau$ effectively be-
come negligible and/or statistically insignificant. The coefficients $a_\tau$ and $b_\tau$ can be modeled us-
ing an exponential filter approach (e.g., Al-Murib et al., 2019); here, as explained below, we esti-
mate the coefficients directly. At a large time lag, the influence of the time-lagged temperature
term in Equation 10 becomes negligible and drops out; hence Equation 11 effectively represents
$T_w$ as a function of time lagged $T_a$ and river discharge.
The discussion above suggests that linear regression models have a basis in the underlying physi-
cal dynamics (see also Mohseni & Stefan, 1999). However, a number of assumptions and ap-
proximations must be made to represent the 1D ADE as a linear model. Factors such as wind,
evaporation, time or spatial variation in parameters and heating terms, and alterations in depth
are only approximately represented by $T_w$ and Q. Moreover, depending on conditions, different
terms (e.g., depth, heat flux, and velocity) may contribute in varying degrees to the overall heat
balance. Thus, a linearized representation of average conditions during a particular season may
work less well under unusual or extreme conditions.



## 2.4 Statistical Model

Statistical models are often used to interpret and predict $T_w$ patterns, using a number of different regressions, statistical approaches, or machine learning (e.g., Benyahya et al., 2007, Zhu et al., 2018). Within the Pacific Northwest, many studies have developed statistical regression models which use $T_a$ and sometimes also river discharge $Q$ to model measured $T_w$ (Moore 1967; Donato, 2002; Bottom et al., 2011; Mayer, 2012). Such models are simple and run quickly, enabling evaluation of time-periods for which in-situ measurements are unavailable and allowing interpretation of primary forcing factors.

We employ a stochastic modeling approach (e.g., Caissie et al., 1998; Benyaha et al., 2007) in which the dependent variable (water temperature $T_w$) and the independent variables (air temperature $T_A$ and river discharge $Q$) are decomposed into a long term climatological average and a time varying component. A similar approach has also been applied to the Columbia River (Scott, 2020; Scott et al., 2022). For a generic variable $X(t)$ measured daily, the climatological average is defined as,

$$\overline{X(t)} = \frac{1}{y_2 - y_1 + 1} \int_{y_1}^{y_2} \int_{-T/2}^{T/2} X(t) dt \, dy, \tag{12}$$

where $T = 30$ days, $t$ is the integer number of days since the start of the year, $y_1$ is the beginning year of the time series (e.g., 1881), $y_2$ is the end year (e.g., 1890), and the overbar represents the climatological average. The number of years in the average should be long enough to capture natural variability, but short enough to be statistically stationary (i.e., not overly influenced by land use changes or climate change). The 95% uncertainty in the climatological average is given by $\frac{t_* \sigma}{\sqrt{N}}$, where $t_* = 1.96$ for a large sample size $N$, and $\sigma$ is the standard deviation. In practice, the number of years we used to define the climatological average is limited by available data.

The deviation from climatology, caused for example by a heat wave, is defined as:

$$X'(t) = X(t) - \overline{X(t)} \tag{13}$$

The climatological average for water temperature, $\overline{T_w(t)}$, is a good first approximation for conditions at any given year-day, and correctly estimates daily $T_w$ in Portland to within a root-mean-square-error (RMSE) of ~1.5 to 2 °C. For a model to have predictive and explanatory power, it must exhibit an RMSE significantly less than this climatological average. Present-day numerical models typically fulfill this criterion and have an RMSE <1°C (Dugdale et al., 2017). To obtain comparable error statistics, we rewrite Equation 11 in terms of deviations of $T_w$ from climatology, and form the following basis function:

$$T_w'(t) = \sum_{\tau=0}^{\tau=j} a_\tau(t-\tau) T_a'(t-\tau) + \sum_{\tau=0}^{\tau=j} b_\tau(t-\tau) Q'(t-\tau) + C, \tag{14}$$

where the prime indicates a deviation from climatology and other terms are as defined in Equation 11. Based on experimentation, we use daily $T_a'$ out to two weeks. Thereafter, we use average $T_a'$, to obtain a statistically significant correlation. A 15 day average is used for day 15– 30, and 30 day averages are used thereafter, up to 6 months. Similarly, river discharge $Q'$ is averaged




using a 10 day average for day 1– 10, a 20 day average for day 11– 30, and – a 30 day average
thereafter.
A total of 8 statistical models are developed, based on data from the 19[th] century (1881– 1890),
mid-20[th] century (1941– 1952), and modern period (2000– 2015) (see Table 2).  These periods
were chosen based on available data; they approximate (nearly) pre-development conditions, pre-
flood control conditions, and modern conditions. With-in each model, we further divide the year
into a summer sub-model (July– September), a winter sub-model (January– March) and an an-
nual model, based on all available data.  Experimentation was used to obtain the optimal winter
and summer models.  For example, including June or October into the summer model signifi-
cantly reduced goodness of fit and the statistical influence of river discharge, consistent with the
observation that the horizontal temperature gradient is largest from July to September (Figure
2b).  Through experimentation, we also determined that discharge only produces a statistically
significant effect for summertime models based on 1941– 1952 and 2000– 2015 data. This result
is consistent with previous studies (e.g., Isaak et al., 2012) and with estimates of $\frac{\partial T_w}{\partial x}$ (section
2.3, Figure 2) which suggests that discharge effects are most prominent in summer.
Results show that the best-fit coefficients generally decrease in magnitude as $T_a$ (Figure 3a,b,c)
and river discharge (Figure 3d) are lagged backwards in time. Further, the decorrelation structure
is different for the 19[th], mid-20[th], and 21[st] century models (Figure 3); hence, for the same forcing,
these statistical models will produce a different output.  Statistically significant coefficients are
found at up to 3 month lag in the 1880s model, and 4 months in the others.




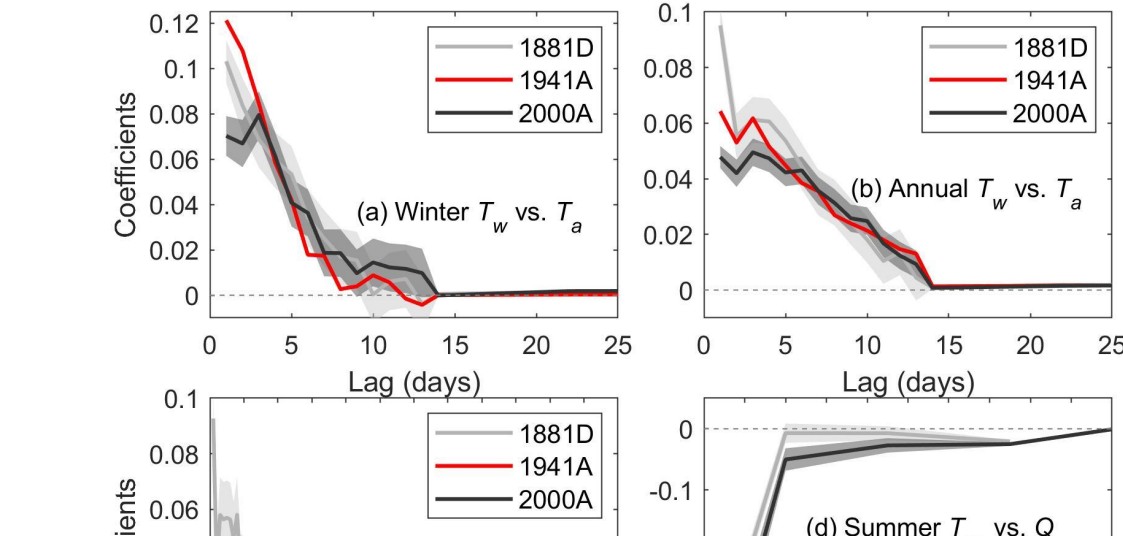


*Figure 3: Coefficients for statistical model vs time lag for (a) air temperature ($T_a$) in the winter model (Nov– Mar); (b) $T_a$ in the annual model (all months); (c) $T_a$ in the summer model (July– Sept) and (d) discharge Q in the summer model (July– Sept). The 1881 model is calibrated to 1881– 1890 $T_w$ data, the 1941 model is calibrated to 1941– 1952 $T_w$ data, and the 2000 model is calibrated to 2000– 2015 $T_w$ data. The letter denotes whether $T_a$ data was sourced from Downtown Portland (D) or from the Airport (A). Similar results are found for the model based on Vancouver air temperature data (not shown). No statistically significant effect of river discharge was found for winter or annual models, and the 1880s summer model, and are not shown.*

Each statistical model produces an estimate of $T_w$ over the period of record of its underlying $T_a$ record (Table 2; data available as supplemental information). Based on these time series, a composite estimate of modeled $T_w$ was produced, as follows. First, for each station, estimates from the two seasonal sub-models were combined, with annual sub-model results used at other times. To avoid (typically small) discontinuities between sub-models, a 15-day linear relaxation period between sub-model start and stop times was applied. Next, a composite estimate for $T_w$ was made for the 1850– 2020 period, using the best available meteorological measurements and statistical models. Vancouver measurements were used pre-1868, downtown Portland from 1874 to 1939, and the Portland Airport data thereafter. Water temperature estimated from Eola $T_a$ measurements were used to fill the 1870– 1874 period. A compromise was required when deciding which era of model to use in the composite, since there is no clear delineation between pre and




post-reservoir conditions, or between a nearly natural and substantially altered landscape. The
mid-20th century calibration, representing pre-reservoir, post-landscape change conditions, was
applied to the 1900– 1960 period; thereafter, we assume modern flood control, and applied the
modern calibration.  Pre-1900 estimates used the calibration based on 1880s data, except for the
Vancouver period (1850– 1868), which used the mid-20th century model because there was no
19th century model. The validity of the composite modeled $T_w$ is assessed, to the extent possible,
through comparison with in-situ measurements (see Results).
Uncertainty was assessed by evaluating the root-mean-square error (RSME) between the compo-
site model estimate and measurements, and comparing against the RMSE found using climatol-
ogy. The uncertainty in each temperature estimate was assessed using a Monte Carlo approach.
Two thousand possible ensembles of the model coefficients were created, under the assumption
that coefficient uncertainty was normally distributed. The 95th percentile of the resulting spread
of solutions is reported.

## 561   3.0 Results and Discussion

### 562   3.1 Model Assessment

Time-series comparisons of water temperature (Figure 4) and statistical evaluations (Table 2)
confirm that the statistical model reproduces reasonably well year-to-year differences in $T_w$ and
weekly-monthly perturbations caused by persistent warm/cold weather. Some synoptic scale
events of less than a week are only partially captured, possibly because of factors not included in
the model (such as cloud cover, wind, or depth changes due to backwater from the Columbia
River; see also Wagner et al., 2011) and the tendency of statistical models to underestimate ex-
tremes. The RMSE between the measured and modeled daily minimum $T_w$ varies from 0.87 to
1.1 ºC for the annual model, with RMSE as low as 0.53 ºC and 0.72 ºC for the summertime and
wintertime models, respectively (Table 2).  Results are less good using Eola, a weather station
which is located ~70km from Portland and may imperfectly represent local meteorological forc-
ing.  On a monthly averaged scale, RMSE varies from ~0.3 to 0.9 ºC, with the best agreement
obtained during the modern period and the summertime sub-models (Table 2).
Our statistical model results compare favorably with numerical models; the RMSE at Portland
for a calibrated numerical model based on measurements from April through September 2002
was 0.43 ºC (Berger et al., 2004), compared to 0.52 ºC for our model over the same period.  Sim-
ilarly, the model performs significantly better than estimates based on $T_w$ climatology, which we
calculate has a root-mean-square error (RMSE) of 1.86, 1.46, and 1.43 ºC for the 1881– 1890,
1941– 1952, and 2000– 2015 calibration periods, respectively.  We conclude that the statistical
model accurately represents the most important factors affecting $T_w$, as long as the underlying
measurements driving the model are reasonably accurate.
Modeled $T_w$ estimates based on different $T_a$ data series (Table 2)   compare well with each other,
with similar averages and variability. During their period of overlap from 1940– 1973, modeled
water temperatures are slightly larger (0.08 ºC) using the airport model (1941A) than the down-
town Portland model (1941D). Similarly, the Vancouver model (1941V model) is 0.02 ºC lower
than the airport model (1941A) between 1940 and 1965.  For the same periods, the daily RMSE





between the 1941A model $T_w$ and the 1941D and 1941V models is 0.29 ℃ and 0.32 ℃, respec-
tively. For the 1896– 1965 period, the 1941D and 1941V models show a mean difference of
0.06 ℃ (Vancouver larger), and an RMSE of 0.37 ℃. These observations provide an order of
magnitude estimate of the aggregate influence of input data and model variability on uncertainty,
whether caused by spatial variations in $T_a$, differences in the statistical coefficients, or instrumen-
tal measurement uncertainty. The consistency and small RMSE between model results improves
our confidence in results.

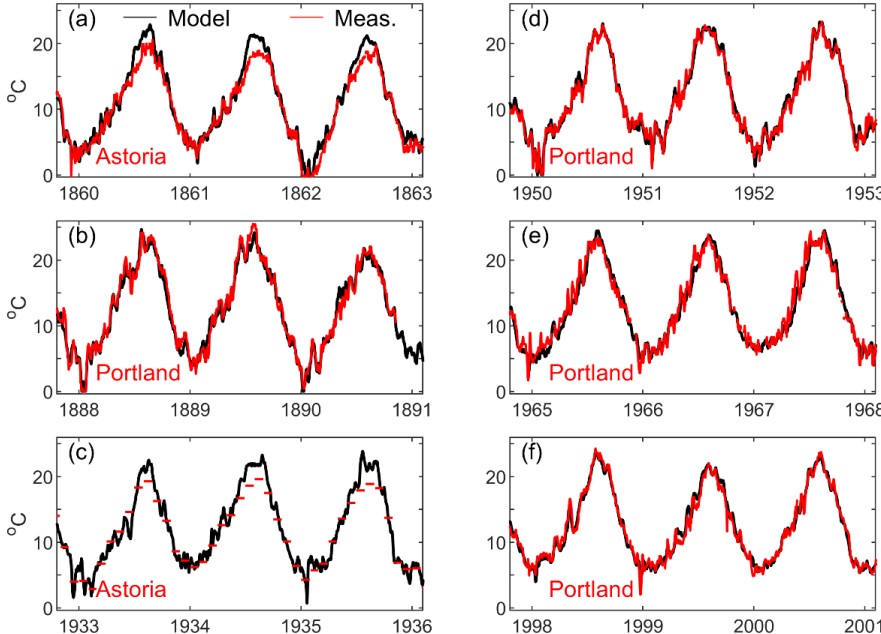


*Figure 4: Comparison of modeled and measured $T_w$ for six periods of three years. The composite Port-*
*land $T_w$ is used in (b), (d), (e) and (f), while Astoria measurements are used in (a) and (c). Only monthly*
*averages of $T_w$ are available at Astoria from 1925 to 1940 and 1943– 1948 (see Table 1).*
One of the factors driving the larger RMSE in the historical model is the larger overall system
variance measured for $T_w$. The typical distribution of $T_a$ anomalies from the climatological mean
has remained stationary between different time periods, and the standard deviation is nearly the
same (within ~5%; Figure 5). However, between the 1880s and the 2000– 2015 period, the dis-
tribution of measured $T_w$ anomalies markedly contracted– , and the standard deviation decreased
from 1.86 to 1.42 ℃ (Figure 5). Since the distribution of $T_a$ anomalies remained similar, a likely
explanation for the decreased variance in $T_w$ is anthropogenic change to the local environment
(e.g., flow regulation, landscape changes, system deepening), rather than changes in meteorolog-
ical forcing (see below for further discussion).





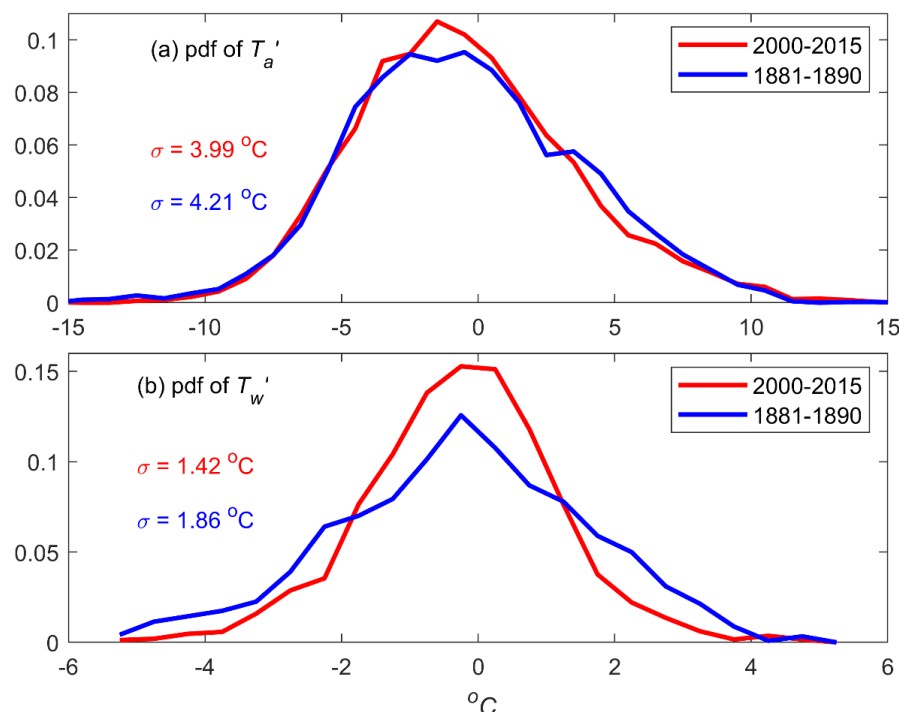


*Figure 5: The distribution of $T_a$ and $T_w$ around the 30d climatological mean for the 1881– 1890 and 2000– 2015 periods.*


## 3.2 Water Temperature Changes in lower Willamette

Model results and measurements show that water temperatures have increased steadily since the 1800s. Increases are observed at all times of the year (Figure 6), leading to an increase in annu-ally averaged $T_w$ of $1.1 \pm 0.2$ °C/century (Figure 7). The largest increase occurred in winter; dur-ing January– February, the trend in average $T_w$ is $1.3 \pm 0.3$ °C/century (Figure 6a). Similarly, the minimum annual temperature is increasing quickly, at $1.8 \pm 0.5$ °C/century (Figure 7b). The smallest bi-monthly averaged trends occur in late spring, during May– June ($0.82 \pm 0.3$ °C/cen-tury trend; Figure 6d). Maximum summer temperatures are trending upwards at ~$0.9 \pm 0.3$ °C/century (Figure 7c), smaller than the annual average. Overall, model results (grey) track avail-able in-situ measurements (red) well, except for some months during periods of lesser data qual-ity in the 1960s– 1970s (Figure 6 & 7). Therefore, modeled and measured trends are consistent, increasing confidence in results.





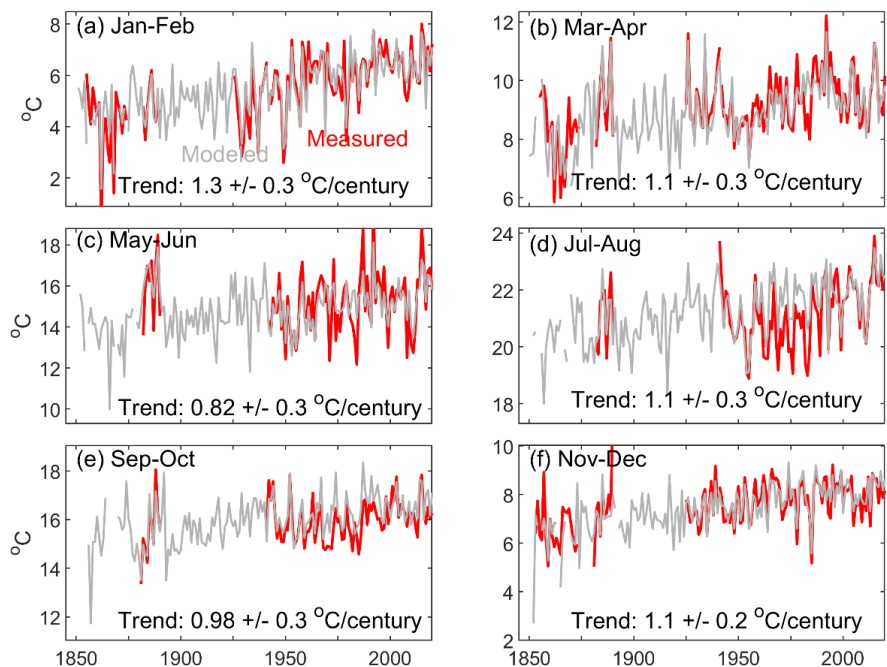

*Figure 6: Seasonal trends in water level, averaged over two month water periods. A correlation is found between measurements (red) and model results (grey). Trends and 95% confidence interval based on a linear regression to model results, 1850– 2020. November– April data from 1854– 1876 from Astoria, Oregon (see Talke et al., 2020). Note different limits on the y-axis.*

No single event or individual system perturbation appears to be causing trends, as there are no step-function changes or inflection points in $T_w$ trends (Figure 6 & 7). Instead, an upwards tendency in $T_w$ is interspersed by large year-to-year variability. In the modern system, the largest interannual variation occurs during the spring period (May– June), with swings of ~5°C observed between years (Figure 6). The late summer and autumn season (September– December) is least variable (order ~2 °C variability between years). Historically, greater year-to-year fluctuations occurred in both measurements and model results, particularly during the cooler half of the year (November– April). Cool-season measurements at Astoria (1854– 1876) between November and April confirm this variability, and track modeled results despite its location on the Columbia River (see e.g. Figure 4a and 4c). The correspondence occurs because during late fall and winter, proportionally more water in the lower Columbia is sourced from coastal tributaries, especially the Willemette, than during other times of year (see Naik and Jay, 2011 and Hudson et al., 2017).

Both climatic factors and system changes drive the reduction in interannual variability in $T_w$. Storage reservoirs, with a large thermal inertia, are one factor (see section 3.3). The change from a multi-braided, shallow channel to a single, deeper channel is also likely influential. Another reason for historical $T_w$ variability in winter was the occasional occurrence of deep freezes that





no longer occur. During the 1861– 62 and 1867– 1868 winters, for example, air temperatures re-
mained below 0 ºC for 32 and 31 days, respectively, and newspapers recorded ice-skating on the
lower Willamette River. Navigation in Portland Harbor was halted or hindered by ice from New
Year's Day until mid-March, 1862. No 20th century winter matched the duration or severity of
these events, though 18– 19 freezing days (maximum below 0 ºC) were recorded in 1915– 16,
1929– 30, and 1949– 50. In 1979, air temperatures remained below 0 ºC for a total of 14 days;
since 1980, no winter has produced more than 9 sub-freezing days. Because some historical win-
ters were mild (e.g., only one freezing day was recorded in 1862– 1863), historical water temper-
atures in winter were much more variable than today.

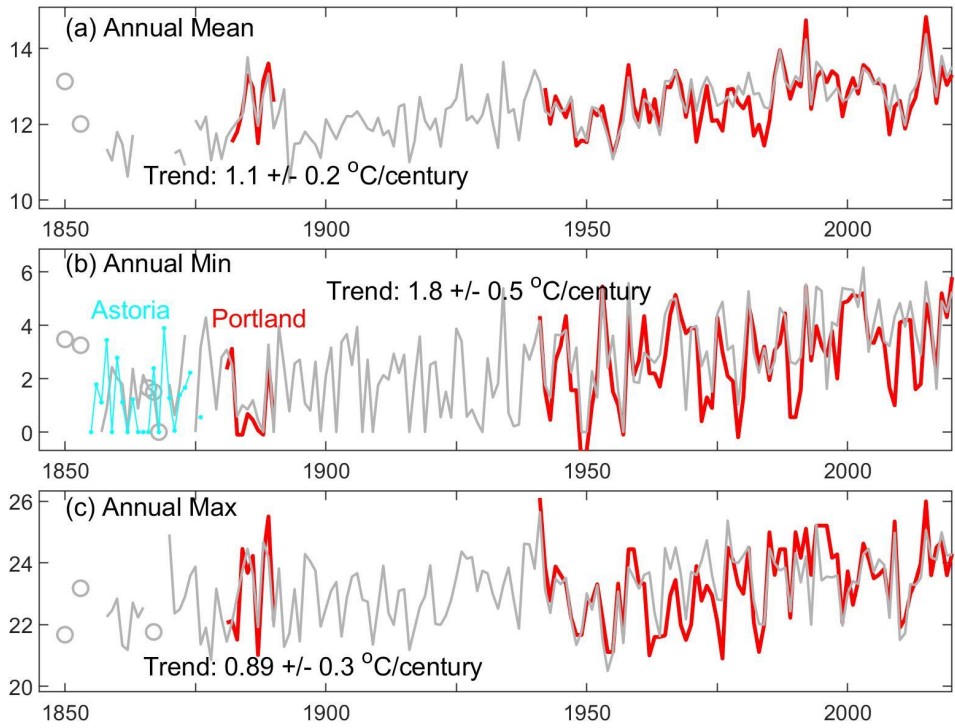


*Figure 7: Time rate of change of annual mean, annual minimum, and annual maximum $T_w$. Grey de-*
*notes model data, red denotes data from Portland region, and cyan denotes $T_w$ measurements in Astoria*
*(annual minimum only). The trend is calculated by regression fit to the 1850– 2015 period. Evaluation is*
*based on daily minimum $T_w$ (see section 2). Years in the 1850s and 1860s without sufficient model data*
*are excluded.*
Results suggest that $T_w$ has always exceeded a threshold of 20 ºC during summer for ~15– 90
days, even during the 1800s (Figures 4, 7c, 8 and 9). A spaghetti plot of all available in-situ data
shows that most $T_w$ measurements exceed the 20 ºC threshold in July and August (Figure 8).





Peak temperatures typically occur during July or August, with no trend in timing observed (Fig-
ure 8, 9). The timing meteorological heat heat waves within a summer—which appears to be ran-
dom—drives the timing of the peak. During some cool summers historically (e.g., 1949; see Fig-
ure 8), temperatures sometimes temporarily dipped below 20 ºC during summer, and remained
above the threshold for less than 2 months. In other years, $T_w$ reaches a peak of 25– 26 ºC, and
water temperatures remain above the biologically important 20 ºC threshold from June to Sep-
tember. During the hot, low river discharge summers of 1889and 2015 (Figure 8), water temper-
atures exceeded 20 ºC for 91 and 95 days, respectively. The biggest difference, in line with other
observations, is that $T_w$ was more variable during the summer of 1889 than in 2015.

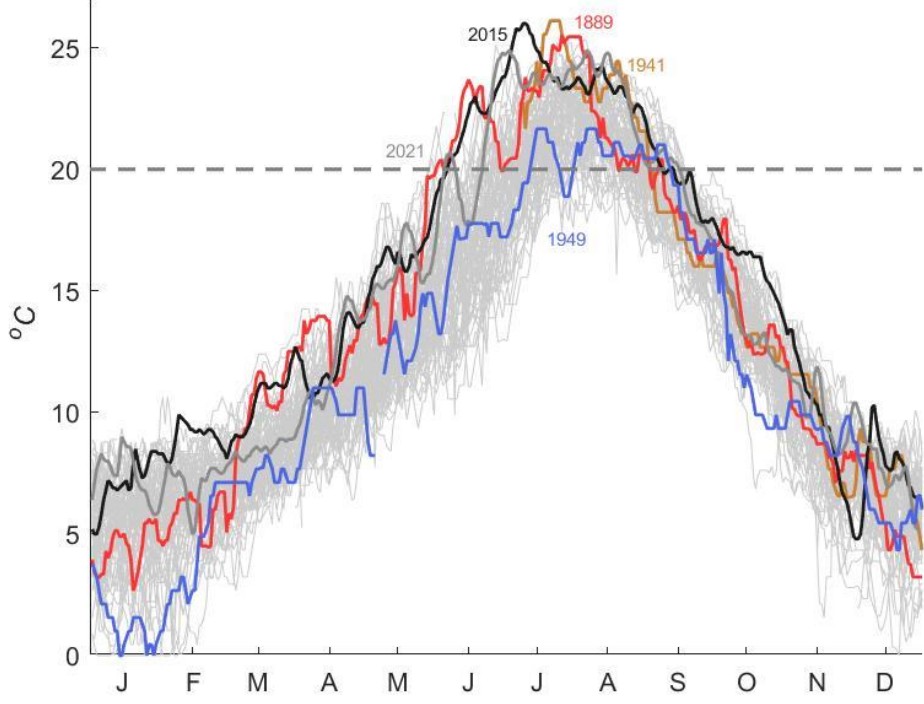


*Figure 8: Spaghetti plot of all measured $T_w$ data from between 1881– 1890 and 1941– 2021. Five years*
*(1889, 1941, 1949, 2015, and 2021) are colored as labeled. Time is labeled at the midpoint of each*
*month.*





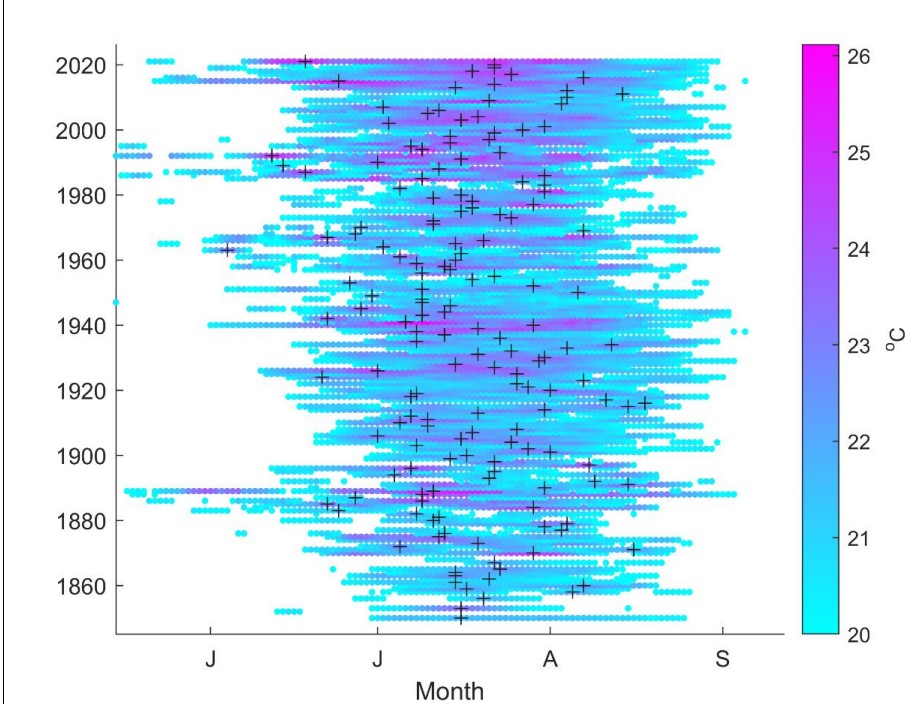


*Figure 9: Summertime $T_w$ values in the Willamette River that exceed a threshold of 20 °C, from 1850 to*
*2021. The instrumental record is used between 1881 and 1890 and 1941 to 2021, and the remainder is*
*infilled with modeled $T_w$. Crosses denote the time of the peak annual $T_w$. Missing air temperature data*
*precluded peak estimates for 1851– 1852, 1854– 1855, 1857, 1866, and 1868– 1869 (see supplemental*
*data). Time is labeled at the midpoint of each month.*

Summers with persistently elevated temperatures occur more often today, even though warm wa-
ters occurred in some historical years (Figures 8 & 9). Between 1881– 1890, measurements
show that the 7-day average temperature exceeded the effective regulatory limit of 20.3°C (see
Introduction) between 11– 80 days, with an average of 42 days. For the 2000– 2021 period, the
range was 35– 92 days of exceedances, with an average of 63 days (2 months). The more con-
sistently warm summer water temperatures help explain the observed upward trend in $T_w$ (Figure
7). Interannual variability has also decreased, due in part to decreased sensitivity to synoptic
(weather) related changes. Evaluated using a 10 year-average, the number of days per year that
exceed 20 °C increased by roughly ~50% between 1850 and 2020, from around 40 d yr$^{-1}$ to more
than 60 d yr$^{-1}$ (Figure 10), an increase of ~20 d. The threshold of 22 °C was exceeded relatively
rarely in the 1800s (<5 days per year), but is now exceeded nearly 40 days per year. Before
about 1960, there was more variability between decades than at present.

The number of cold-water days in winter has declined precipitously as overall temperatures have
warmed (Figure 10a). Water temperatures are now rarely below 4 °C, compared to about 25 d



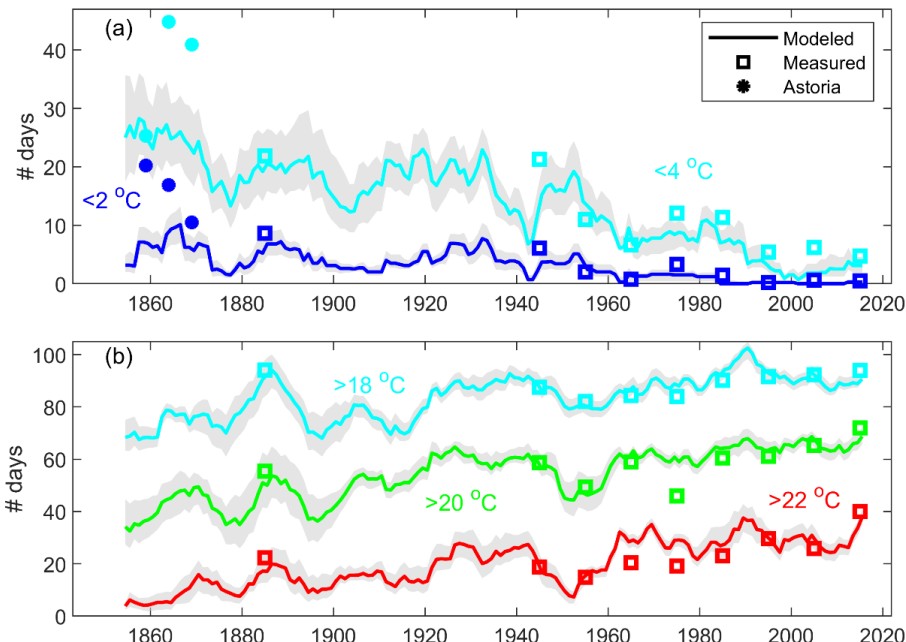

697

*Figure 10: Comparison of the modeled and measured number of days per year from 1850 to 2020 that*
*$T_w$ is (a) below a threshold of 2°C and 4°C and (b) above thresholds of 18°C, 20°C, and 22°C. Square sym-*
*bols denote the 10-year average based on measurements, while the solid line is a running 10 year aver-*
*age of modeled $T_w$. Measurements based primarily on bias corrected upstream gauges (1962, 1983–*
*1984) are excluded. Grey shading is the 95% confidence interval, based on resampling model coefficients*
*using a Monte-Carlo based technique. Wintertime measurements from Astoria (1854– 1876) included in*
*(a).*

per year in the mid-1800s. Similarly, near freezing temperatures (below 2 °C) were common in
the 1800s (up to 10 d yr$^{-1}$), but almost never occur now. While an increase in winter water tem-
peratures has received much less attention than summer-time trends, this shift is also ecologically
important (e.g., Webb & Weber, 1993; Caissie, 2006). For example, cold water events and win-
tertime conditions influence the survivability and recruitment of fish by altering their biotic inter-
actions, habitat use, physical condition, feeding rates, and community structure (see reviews by
Hurst 2007; Brown et al., 2011; Weber et al., 2013). It is also possible that historical wintertime
conditions, such as the deep freezes discussed above, provided some protection against non-na-
tive plants and fauna that thrive in warmer waters.

## 3.3 Interpretation of water temperature changes

In general, seasonal patterns of measured $T_w$ and shifts between 19[th] and 21[st] century data are
consistent with measurements of $T_a$, with some slight variations in timing and magnitude (Figure
11). Measurements in Portland indicate that the daily maximum air temperatures ($T_a$) increased



by 1.3 ºC between the 1875– 1904 and 1991– 2020 periods (Figure 11b), consistent with warm-
ing trends of 0.5– 2 ºC per century at 100+ stations throughout the Pacific Northwest (Mote et
al., 2003) and an average increase of ~1.1 ºC since 1900 (Mote et al., 2019). The smallest in-
creases in Portland $T_a$ occur in spring (April– June) and in late fall (November– December), and
the largest occur in January– February and July– October, again consistent with $T_a$ trends in the
Maritime Pacific Northwest (Mote, 2003). Within Portland, the large summertime increase may
be influenced by the urban heat Island effect (e.g., Voelkel et al., 2018). However, the city has
been relatively urbanized (cleared of forest) since the beginning of the time series, and $T_a$ meas-
urements have primarily occurred by either the Willamette or Columbia River, both reasons that
changes in temperature bias caused by infrastructure may be relatively small. Moreover, the dis-
tribution of air temperatures around the climatological mean has remained virtually unchanged
(Figure 5). Given the long history of Portland and later the Airport as the primary regional meas-
urement station, and the consistency of trends with the regional average (e.g., Mote et al., 2019),
we conclude that the $T_a$ measurements are reasonably representative of regional climate patterns.
Average air temperatures during the 1881– 1890 calibration period (during the Signal Service $T_w$
measurements) are only 0.4 ºC cooler than the 2000– 2015 calibration period (Figure 11d), mark-
edly lower than the 1.3 ºC difference between the 30y climatological averages (Figure 11b). A
possible reason is that pre-1888 measurements may not have been properly sheltered (Mote
2003). However, comparison with $T_w$ measurements (compare Figure 11c with 11e) suggests
that air and water temperature patterns during this decade were similar and warmer than previous
and subsequent decades. For example, both springtime $T_a$ and $T_w$ measurements in the 1880s
were higher than instrumental measurements from the 2000– 2015 period. The correspondence
between $T_a$ and $T_w$ measurements in the 1880s increases confidence that measurements indicate a
real climate signal, possibly caused by decadal fluctuations in climate (e.g., Peterson & Kinkel,
2001), rather than an instrumental artifact.

### 3.3.1 Causes of $T_w$ Change

We next approximate the magnitude of factors causing $T_w$ change using a series of sensitivity
studies. These experiments provide an order-of-magnitude assessment of how sensitive the sys-
tem is to changed coefficients or input data. We evaluate:

1. Integrated system changes. By applying the same input data to models from different
751       time periods, we explore how the system response has changed to the same perturbations.
752       River flow and $T_a$ data from 2000– 2020 are used.
2. The effects of climate change. The climatological $T_a$ increase since the 19th century in
754       Portland is applied (Figure 11b), while river flow and the statistical model are kept the
755       same.
3. The effect of water resources management. The change in the river hydrograph (Figure
757       2a) is applied, with the system coefficients and $T_a$ held constant.






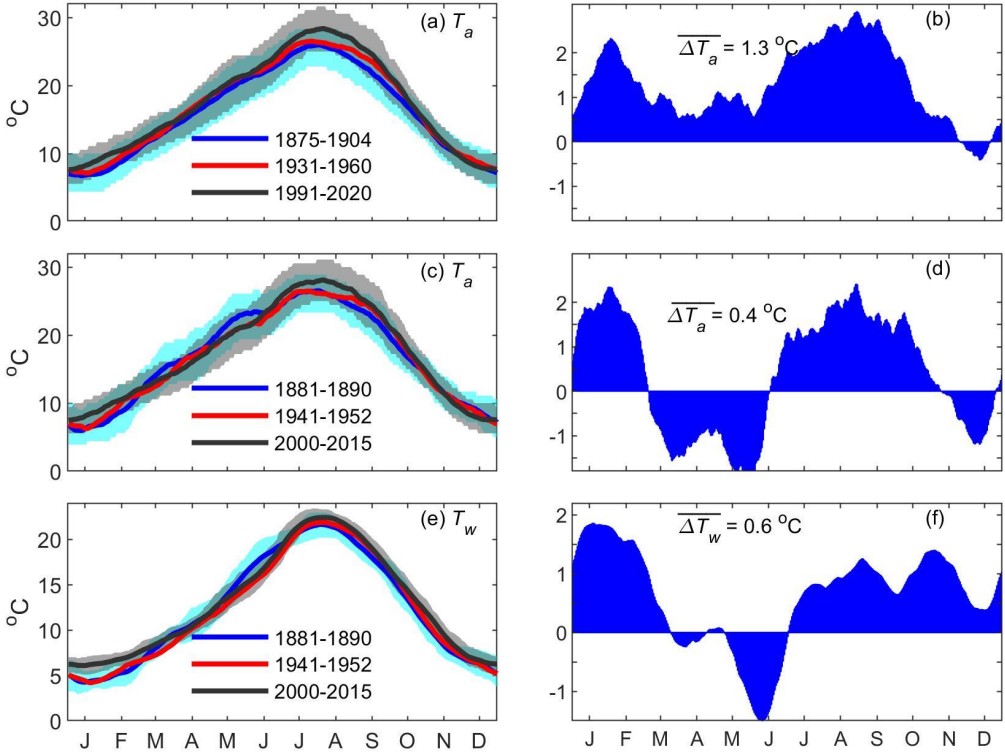


*Figure 11: $T_a$ and $T_w$ climatology in Portland (a,c,e) and the difference between the modern (1991– 2020)*
*and historical period (1881– 1890) in (b,d,e). Climatology is determined using a 30d moving average;*
*shading denotes the 25th and 75th percentile of the measurements. A 30 year average is used in (a); the*
*time periods for (c) and (e) are determined by the time period used to calibrate the $T_w$ model. The tick*
*marks on the x-axis denote the middle of each month. The average $T_w$ difference between the modern*
*and earliest period is provided in (b,d,e).*

Model results confirm that changes in $T_a$ (driven by climate change) are the most significant fac-
tor in long-term increases in $T_w$, with system changes an additional important contributor during
the cool season (Figure 12). Seasonally, changes to $T_a$ between the 1875– 1904 and 1991– 2020
periods dominate the modeled trends in $T_w$ during summer and early fall (July– October) and in
late winter (Figure 12). Averaged over a year, a total increase in $T_w$ of 0.81 ± 0.25 °C is corre-
lated to $T_a$ changes. A maximum climate-induced change of ~1.7 ± 0.3 °C occurs in September.
Climate shifts produce a lesser shift of 0.5– 0.6 °C increase in $T_w$ in spring (late March to June),
and little change occurs in December, consistent with air-temperature climatology (compare Fig-
ure 11 and 12). Interestingly, uncertainty in the air temperature contribution is driven by the in-
herent 95% confidence in the air temperature climatology, which is ±0.22 °C, rather than uncer-



tainty in the model coefficients. Moreover, modeled $T_w$ changes are robust to any small system-
atic biases in $T_a$; if the average change in $T_a$ is reduced by 0.5 ºC, the average $T_w$ only decreases
by ~0.3 ºC. Hence, we conclude that changes to the meteorological heat-balance (as represented
by $T_a$) are the major cause of increasing $T_w$. Climate models also suggest that future summertime
$T_w$ in the Pacific Northwest will increase much more than other seasons, consistent with our re-
sults (Ficklen et al., 2014).
System changes (as estimated by changing regression coefficients) between the 1940s and today
cause a $T_w$ increase of ~0.5– 0.6 ºC from November– May, dropping to a statistically insignifi-
cant amount from late June to early October. Averaged over a year, the total increase in $T_w$
caused by system change is $0.34 \pm 0.12$ ºC. The observed seasonal shifts are consistent with an
increased thermal inertia caused by the reservoir system, as also discussed elsewhere (see e.g.
Webb & Weber, 1993; Caissie 2006; Olden and Naiman, 2010). Effectively, heating or cooling
from many months ago still influences $T_w$ in the modern system, tending to elevate wintertime
and depress summertime temperatures (see discussion for other influences). In the statistical
model, we find that monthly averaged $T_a$ exerts a statistical influence on $T_w$ for 4 months, com-
pared to 3 months historically (not shown). The coefficient magnitudes at 2– 4 months lag are
also larger today, at ~0.0025 º$T_w$/º$T_a$ per day (modern) vs. ~0.0017 º$T_w$/º$T_a$ per day (1940s; an-
nual model). The other significant change in the modern model is a lessened sensitivity to syn-
optic weather patterns, as observed by smaller coefficients at <7 days lag (Figure 3) and less var-
iance (Figure 5). Both the decreased sensitivity and the longer system memory in the modern
system affect the modeled $T_w$, leading to the changed pattern of $T_w$ responses to atmospheric
forcing.
Changes in average river flow exert a minor influence on annually averaged $T_w$, but are im-
portant during late summer. During July, a slight increase in $T_w$ is observed from changed river
flow. In August and especially September, the decrease in $T_w$ caused by increased flow releases,
-0.27 ºC and -0.56 ºC are significant. Thus, the release of water from reservoirs late in the sum-
mer counteracts, to some extent, the effects of increased air temperatures. During other times of
year, no statistically significant modeled correlation between $Q$ and $T_w$ was found, likely because
the average $T_w$ gradient in the mainstem Willamette River is small (Figure 2b). While river flow
may be important in winter during times of large positive or negative temperature gradients,
these changes are likely transient and a process-based model would be required to capture it.
The net effect of summertime changes in river flow on the annual average is small: A total de-
crease in annually averaged $T_w$ of ~0.05 ºC is estimated.

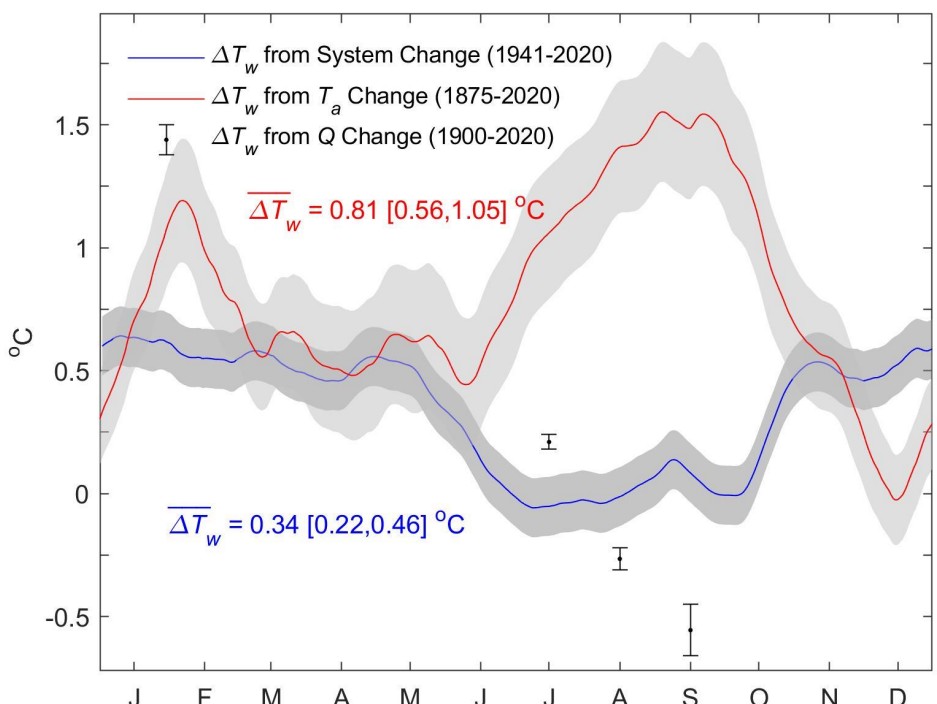


*Figure 12: Estimated $T_w$ changes caused by $T_a$ (climate change), system changes (i.e., differences be-
tween the parameters of the modern and historic models), and discharge changes (July– September). A
positive value indicates an increase over time. System changes are based on taking the difference in esti-
mated $T_w$ obtained over the 2000– 2020 time period using the 1940s era model (model 1941A) and the
modern era model (model 2000A). The influence of increased solar heating (climate change) is estimated
by differencing the $T_a$ and $T_w$ values obtained with the 1941A model using the 1875– 1904 and 1991–
2020 $T_a$ climatologies (Figure 11). Shading includes the both the 95% uncertainty in the mean climatol-
ogy and the 95% uncertainty in the coefficients. The July– September change in monthly averaged Tw
produced by discharge is obtained by comparing the pre-reservoir (1901– 1940) and the modern (1981–
2020) hydrographs (Figure2a) into the statistical model, with $T_a$ forcing held constant. The error bars de-
note the difference in the 1941A and 2000A model estimates.*


Overall, the sum of estimated temperature changes caused by climate, system, and water man-
agement changes since ~1900 (~1.1 ± 0.3 °C; Figure 12) is consistent with the overall long-term
trends in $T_w$ of 1.1 ± 0.2 °C per century (Figure 7a). Thus, we conclude that ongoing climate
change is the primary cause of increased temperatures, with system changes an important con-
tributor. We note again that we cannot discern the influence of individual factors such as
changed shading, river depth, storage, or snow pack, nor can we assess coupled, nonlinear





changes. For example, changes to river flow may in part be caused by climate change, and altera-
tions in $T_a$ may in part be influenced by urbanization or deforestation. Nonetheless, the results
provide insights into the causes of $T_w$ change and why some parts of the year are subject to larger
upward trends than others, over secular timescales.
## 4.0 Discussion
The observed annual trend in $T_w$ of 1.1 ± 0.2 °C/ century in the lower Willamette River is similar
to the magnitude of change observed or estimated in the few studies available over similar time
scales. For example, Moatar and Gailhard (2006) estimated a 0.8 °C increase in the Loire since
1881, Webb and Noblis (2007) estimated a change of 1.4– 1.7 °C on Austrian rivers since ~1900,
and Scott (2020) estimated a trend of 1.3 °C/ century for the nearby Columbia River over the past
170 years (see also Scott et al., 2022). Similar to our results, studies also often highlight that the
seasonal distribution of changes of $T_w$ is unequal (e.g.,,Webb and Noblis, 2007). Consistent with
our results, studies from the Pacific Northwest suggest that climate change is driving $T_w$ trends
over recent decades (Isaak et al., 2012). Future climate change will continue to drive trends,
with the largest increases in summer (Caldwell et al., 2013; Ficklin et al., 2014). But, our results
suggest that system changes have altered the response of $T_w$ to climate change, and in particular
extremes, as explored below.
Both measurements (e.g., Figure 5) and the statistical model coefficients for $T_a$ (Figure 3) sug-
gest that the sensitivity of $T_w$ to short-term meteorological forcing has decreased over time. A
major cause is the reservoir system, which is known to decrease $T_w$ variability in the Willamette
on 1– 8 day time scales (Steel and Lange, 2007). At short time lags of 0–5 days, historical model
coefficients are as much as 2–3x larger than modern coefficients (Figure 3). Therefore, a histori-
cal heat wave in $T_a$ was likely to produce a larger change in $T_w$ than today. Simultaneously, the
integrated effect of weather during previous months is more important. At lags of > 2 weeks, co-
efficient magnitudes are ~50% larger in the modern models than historically. Hence, the thermal
memory of the system to $T_a$ anomalies lasting a month or longer is larger. Thermal memory
stems from storage effects, whether from the heat stored in reservoirs (Webb & Weber, 1993;
Caissie, 2006; Olden & Naiman, 2010) or the cooling effects of snow melt and groundwater in
summer, which together are the primary source of water during this period (Brooks et al., 2012).
The net thermal memory has increased, providing a buffering effect that helps explain why both
seasonal and interannual variations in $T_w$ are less pronounced today.
We attribute the decreased sensitivity of $T_w$ to short-term, synoptic weather patterns (< 1 week)
to a system-wide increase in depth, caused by the reservoir system (Rounds, 2007,2010) and by
channelization and depth increases in the river (Sedell & Froggatt, 1984; Gregory et al., 2002a)
A larger depth $d$ decreases the magnitude of the heating term ($\frac{H}{\rho c_p d}$) in Equation (1), leading to
smaller temperature change in the leading order balance $\frac{\partial T_w}{\partial t} = \frac{H}{\rho c_p d}$. This explains the decrease
in model coefficients for small time lags (< 1 week). Reservoirs in the upper watershed increase
the mean depth of the entire system, reducing the overall rate of temperature change but increas-
ing heat storage capacity (Caissie, 2006). Similarly, the transition from a multi-braided stream to
a dredged river with one primary channel also contributes to increased depth, to an unknown ex-
tent. Gravel mining and dredging for the harbor may also have increased depths in the lower



Willamette (see e.g., Jay et al., 2011). These Portland-region depth increases may be offset by a
decrease in backwater effects from the Columbia River, particularly in spring (Helaire et al.,
872 2019).

The changing correlation structure (Figure 3) and the influence of increasing depth has implica-
tions for how climate change effects are observed. At short time scales (<1 week), the decreased
modern sensitivity to air temperature perturbations (Figure 3) implies that depth increases out-
weigh altered $H$ in the heating term. If the correlation structure had remained unchanged, a 1 °C
step increase in $T_a$ would result in a larger short-term perturbation than is currently observed.
Hence, $T_w$ in the modern system has become more resilient to extreme heat waves. The record
breaking heat wave in July 2021, with a high $T_a$ of 46.7 °C, did not cause a record $T_w$. Despite
air temperatures exceeding the previous all-time high by ~5 °C, the daily minimum water tem-
peratures peaked just over 24 °C, in part because the heat wave was shorter than other events.
We conclude that water temperatures are now more influenced by climate-change induced
changes to air temperature climatology and long-time scale patterns, rather than short-term ex-
treme events.
Numerical, process based models run over a smaller time scale provide additional clues to the
factors driving long-term changes. For example, non-reservoir anthropogenic factors were mod-
eled to increase Willamette River water temperatures in Portland by 0.3 ±0.05 °C between June
and October of 2001 (OR DEQ, 2006), primarily due to loss of shading (86%) and secondarily
because of point-source discharges (e.g., from water treatment plants). The same CE2-Qual
model determined a reduction of approximately 0.1 °C for each additional 100 m³/s of river flow-
released into the lower Willamette. This is consistent with our modern statistical model, which
suggests an influence of ~0.07 °C for each extra 100 m³/s of river flow, spread out over several
months via the decorrelation structure (Figure 3d).
River flow effects on $T_w$ are likely driven by the substantial positive summertime $\frac{dT_w}{dx}$ (Figure
2b) during July-September, but are also influenced by the increased velocity and depth caused by
each incremental increase in discharge (see Equation 1). The large increase in September dis-
charge (Figure 2) reduces temperatures by 0.56 °C, a larger amount than in August (Figure 12).
In October, average $\frac{dT_w}{dx}$ becomes small (Figure 2), and our approach is unable to find a statisti-
cally significant influence of river discharge.
Interestingly, the overall system was less sensitive to river flow fluctuations in the 1940s (Figure
3d), and no statistically significant effect was observed in the 1880s. The lack of correlation in
the 1880s may simply reflect imperfect flow estimates (see Jay & Naik, 2011). Nonetheless, it is
possible that the bottomland forests and braided river networks of the historical Willamette River
greatly reduced $\frac{dT_w}{dx}$, velocity, and the advective heating term during summer (Equation 1), pro-
ducing the observed lack of correlation. Mechanisms that might be influential include stream
width changes (e.g., White et al., 2017) and cold groundwater discharges, which is known to oc-
cur in off-main channel alcoves (e.g., Faulkner et al., 2020). During winter, the shallower histori-
cal streams may have contributed to the freezing water temperatures observed during some years
in the 19th century. A process-based retrospective model using historical bathymetry would be
required to further investigate these conjectures, and determine the relative roles of geomorphic
change, ecological change, and the reservoir system on $T_w$.





Since spring $T_a$ values are less changed than summer values (Figure 11), less extra heat is input
at the beginning of the warm season, and warm $T_w$ is not biased early in the modern record. In
the late summer, reservoir releases are tamping $T_w$ values downwards (Figure 12).
The increase in the number of days that temperatures exceed a threshold has been observed in
other river systems (e.g., Markovic et al., 2013) and is projected to continue in the Pacific North-
west (Mantua, 2010). Our observations show that the rate of change is threshold dependent, and
slows as the accumulated number of days above a threshold becomes large. Therefore, the num-
ber of days over 20 ºC (which is already large) is increasing less quickly than the number of 22
ºC days, which occur primarily during mid-summer (Figure 9). Effectively, exceedences of
lower thresholds like 18 ºC and 20 ºC are limited by spring and fall, when temperatures change
quickly. Conversely, in winter, the largest rates of change are observed for larger levels of ex-
ceedance; hence, the number of cold-water days below 4 ºC is decreasing faster than those below
2 ºC.  Average temperatures in Jan-Feb, the period with the coldest temperatures, have increased
from ~0-6 degrees to 5-8 degrees (Figure 6a).  Hence, both the decreased spread in temperatures
(Figure 5) and the increased mean drive the large change in the number of days below 4 ºC.
Compared to historical norms, water temperatures today exhibit less variability, both day-to-day
and between the maximum and minimum (both climatology and daily extrema). A result is that
*temporal refugia*—which we define as time periods in which water temperatures temporarily dip
below biologically important thresholds such as 18 ºC or 20 ºC—are becoming less frequent (see
Figure 9,10).  Hence, while the management practice of selectively releasing river water is suc-
cessfully reducing average temperatures in late summer (Figure 12), it may not be addressing the
decrease in variance (e.g., Figure 5) caused by system changes.  Because some migrating fish
such as steelhead delay migration during warm periods by weeks or months, likely causing in-
creased mortality (e.g., Siegel et al., 2021), the reduced temporal refugia are important to con-
sider (see also Steel et al., 2012).  At Portland, $T_w$ exceeds—and has done so throughout the pe-
riod of record—biologically important thresholds during some part of every year.  However, the
more consistently warm temperatures during summer and the shoulder seasons—as observed by
the increase in the time over 18 ºC and 20 ºC—likely creates a thermal barrier that has implica-
tions for salmon migration (see e.g., Notch et al., 2020). .

## 5.0 Conclusion

In this contribution, we found, digitized, produced, and quality controlled a long $T_w$ record
(1881– 1890, 1941– 2021) for the lower Willamette River in Portland, Oregon.  The in-situ
measurements enabled the development of statistical $T_w$ models based on the 1880s, 1940s, and
modern time periods.  Subsequently, estimates of daily minimum $T_w$ for the years 1850– 2021
are produced using daily measurements of $T_a$ and river discharge. A good comparison between
measurements and models is observed (Table 2), including cool season water temperature meas-
urements (November – April) in the Columbia River Estuary from 1854– 1876.
Water temperatures are increasing throughout the year (average trend of 1.1 ± 0.2 ºC/ century),
with the largest trends observed in winter.  As a result, the number of cold water days per year is
precipitously declining, while the number of days above 20 ºC has increased by an average of
~20 d yr$^{-1}$ (Figure 10).  The primary cause of changed $T_w$ since ~1900 is climate change (0.84
ºC), followed by system changes such as the building of reservoirs, loss of shading, and other



landscape alterations (0.34 °C; Figure 12). Changes and reductions in flow have a generally
smaller influence. Because of a larger heat capacity and greater system depth, the day-to-day var-
iability in $T_w$ has decreased (e.g., Figure 5). Even though average temperatures in summer are
now larger, peak temperatures have changed less. Hence, warm summers marked by low river
flow produced similar peak temperatures in 1889, 1941, and 2015 (Figure 9), and a truly extreme
heat wave in 2021 did not produce record water temperatures, possibly because of its short dura-
tion. The relative suppression of peak $T_w$ has been bought at the expense of daily and interannual
variability; during most times of the year, but particularly in winter, there is less day-to-day vari-
ation than in the 19th century. Climatic induced disturbance events such as freezing rarely occur
anymore. Similarly, temporal refugia—time periods in which $T_w$ dips below biologically im-
portant thresholds—have also decreased (Figures 9 & 10). These system changes may pose a
grave threat to endemic species, should climate-induced changes in $T_w$ continue.

## Data Availability

The water temperature data used in the research is available upon request, and will be uploaded
to a data repository upon acceptance of the manuscript. Meteorological data are available from
the National Centers for Environmental Information (https://www.ncei.noaa.gov/). Pre-1890
Vancouver and Portland records were also obtained from the Midwestern Regional Climate Cen-
ter (https://mrcc.illinois.edu/data_serv/cdmp/cdmp.jsp ). River flow records are obtained from
the US Geological Survey and the sources described in section 2.

## Author Contribution

SAT found and processed archival data, developed the statistical model, analyzed results, pro-
duced figures, and was primary lead on drafting the paper. DAJ developed an earlier version of
the model and assisted with interpretation and paper development. HLD assisted with interpreta-
tion and paper drafts, and helped secure funding.

## Competing Interests

The authors declare that they have no conflict of interest

## Acknowledgements

Funding was provided by Bonneville Power Administration, under Project No. 2002-077-00 with
the Pacific Northwest National Laboratory, and by the US National Science Foundation, CA-
REER Award 1455350 and NSF project 2013280. Margaret McKeon is thanked for her help de-
fining the watershed boundaries in Figure 1, and students at Portland State University are
thanked for helping to digitize and quality assure the 1854-1876, 1881-1890, and 1941-1961 wa-
ter temperature records used in this study.





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
