# Peer review of "Warming of the Willamette River, 1850–present: the effects of climate change and direct human interventions"

_EGUsphere, 2022_

## Author Comment (AC1)

Dear Editors and Reviewers,

We have closely considered the comments of both reviewers and have undertaken a major rewrite of the text to improve our presentation and make the scientific novelty more clear. We thank both reviewers for their comments, which have greatly improved the manuscript. Pending approval from the editors, we will upload the updated manuscript. In the mean-time, we have outlined our edits and include details about specific improvements.

As described in the detailed responses below, the introduction is now more succinct and focusses more clearly on broad relevance and scientific questions. Much of the locally specific material has been moved to either section 2.1 (Methods>Study Area) or to a supplement. The methods section has been reduced by more than 25%, both by careful editing but also by moving supporting information to the supplement. The results section has been streamlined to more clearly focus on a description of results, following scientific convention. The discussion has been reorganized and interpretation and analysis has been moved there from other sections. The reorganization more clearly focuses on interpretation (section 4.1), implications (section 4.2), and limitations (section 4.3). Where requested, additional explanations and specificity have been added. Overall, the text is more than 100 lines shorter.

Through the careful rewrite, the modeling approach and novel contributions are more clearly findable and understandable, and the methods are more succinctly and clearly explained. The concerns of both reviewers concerns about the length and understandability of the manuscript have therefore been addressed.

As described in the detailed responses below, the scientific novelty and broad impacts of the study include the construction of a uniquely long record, the evaluation and explanation of seasonal trends and patterns over secular time scales, and a method for attributing changes to climate and river system changes over a period that pre-dates reservoir construction and other anthropogenic impacts. We thank reviewer 2 for their supportive comments and acknowledgement of the scientific contributions of the paper, and respectfully disagree with reviewer #1 that our manuscript and results lack novelty. However, we acknowledge that many of the novel aspects were implied, and may not have been sufficiently apparent because of the length of the manuscript and its organization. In the revision, we use first person more systematically to take ownership of our specific results and highlight our scientific contributions.

Our model and measurements provide daily measurements of water temperature over a 170 year period in a temperate-zone river that shares many similarities with watersheds worldwide, at an accuracy that is comparable to other statistical and numerical models. We provide both estimates of long term trends, seasonal shifts, and their first order reasons (climate, local river system changes). The in-situ and model results are now available at a data repository. We hope that the editors and reviewers agree that, taken together, our manuscript and results are an achievement worth publishing in HESS.

Best regards, also on behalf of co-authors,

Stefan Talke

**Specific Responses to reviewer Comments**

We thank Reviewer #1 for their comments, which have improved the text. Replies to comments are shown in blue below.

Review of the manuscript: "Warming of the Willamette River, 1850–present: the effects of climate change and direct human interventions" by Talke et al. In its current state, it is far from a scientific research paper, instead a technical report. Most of the content in the discussion of results and conclusions are merely descriptions of their results and a list of general knowledge without any significant novel contribution.

Thank you for your perspective. We acknowledge that we may not have communicated our novel contributions sufficiently. To make sure that our research contributions are clear and not missed by a reader, we have both shortened the paper and more explicitly stated novel aspects, particularly in the introduction, discussion, and conclusion. The results section now focusses more clearly on the description of results, and the discussion has been reorganized and expanded to more clearly communicate the interpretation and implication of results, and the limitations in the modeling and attribution approach.

As outlined below and in the text, there are multiple novel scientific aspects and broader impacts. These scientific findings are more clearly defined and highlighted in the Introduction and in other sections, as detailed in other comments. The scientific novelty and broad impacts include:

- A uniquely long data set, found through archival research and data rescue. This approach can be replicated elsewhere, but requires visiting multiple federal, state, and local archives. Our effort shows this is both possible and productive. Few pre-1900 water temperature data sets are available and this limits the study of global, secular changes to river temperatures (unlike meteorological or oceanographic data). This study is a step towards a goal of describing and understanding such changes. We note that the second reviewer considered the data set to be an important contribution.
- The manuscript provides estimates of water temperature trends and changes to seasonal patterns that go deep into the 19th century. Evaluating changes in the number of warm water days over a threshold over such a long time scale is a novel contribution. Describing how the distribution (standard deviation) of water temperature has changed as a result of river system changes such as deepening and how that impacts means and extremes over a long time scale is an important insight.
- Few if any studies that have investigated how the days below a cold water threshold have declined. Highlighting the causes and consequences of this issue is a novel contribution
- Describing and implementing a methodology for separating out the effects of climate change and river system changes (anthropogenic developments) on water temperature is novel, over such a long time scale. While a process-based numerical model could be used to simulate historical conditions, such efforts are rarely verifiable using data. Our methodology allows us to parse out and attribute changes to individual factors, and our data will enable future numerical studies to further investigate specific mechanisms (e.g.,

the role of shade vs. reservoirs in temperature increases). Reviewer #2 recognized that the attribution analysis was an important part of the paper.

To make the scientific contributions more obvious, we have edited the text to be more succinct, and have moved many methodology details to the supplement. We have also edited the introduction to both provide broader context and explain what our scientific contributions are explicitly (rather than being left implied, as was previously the case for some points).

It is possible that the reviewer might have thought this was a technical report in part because of the obvious importance of the findings to local managers and engineers. Having a local importance does not necessarily imply that scientific results are not relevant elsewhere, and we note here that some specific regulations—e.g., the 20 degree Celsius regulatory limit—are relevant to the rest of the United States and elsewhere, because they represent broad ecological thresholds relevant to other temperate zone rivers. We have rewritten the introduction to make clear that the challenges and changes in the Willamette basin are similar to issues in other temperate basins. We note that many of the references we cite also discuss the importance of thresholds and the stressors affecting temperate-zone rivers, and we have added some more references to make this more clear. Our discussion of regulatory limits was made to motivate the use of a 'days over threshold' approach, as is often done in studies of flooding. There are many interesting aspects of threshold exceedance processes, not least of which that they are often nonlinear—a small change in temperature can lead to a big change in exceedances. The paper therefore focuses on these exceedances from a scientific point of view, but we note that there are strong practical applications (which is not a negative part of the paper, in our opinion).

If the authors manage to rewrite the manuscript some notes about minor details are below:

1. What are exactly system changes mentioned in the short summary? Authors should use some related technical terms.

   Thanks for the comment. We note that the purpose of a short summary is to be understandable to a larger, non-technical audience. We have rewritten the short summary to be more specific, but still accessible to non-specialists. This section now reads:

   *"Approximately 30% of increased water temperature is attributable to changes in the river system, including landscape changes and depth increases from reservoir construction and river channelization. An average annual reduction of about 5% is achieved by summertime river flow regulation. The largest factor driving modeled changes since 1900 are warming air temperatures (nearly 75%). As a result, the number of warm water days has significantly increased, and near-freezing conditions, common historically, no longer occur."*

2. What is the novelty of this paper? Can the author mention some scientific applications including the novel idea?

   During the revision process, we noticed that many novel elements were implied in the introduction, but not explicitly stated. Therefore, we have revised the introduction and

use the first person to point out and take ownership of novel ideas.  A few examples are given below, but more can be found in the introduction and throughout the manuscript:

> (a) First Introduction paragraph was rewritten to take ownership of the following novel idea, which was previously only implied but is now more obvious:

*"In this study, we find, recover and analyze previously forgotten or unused archival $T_w$ records from 1881 onward for the lower Willamette River. These records, which precede most industrialization and modern development in the Pacific Northwest, provide a unique opportunity to discern secular trends, evaluate and attribute causes, and assess the net impact of human activities in a temperate coastal river."*

> (b) The second paragraph has been simplified and now explicitly states that our approach is new and addresses a common existing problem, namely that pre-reservoir temperature evaluation is usually not possible:

*"Because of a lack of in-situ data from pre-reservoir conditions, the cumulative effect of anthropogenic influence is unknown (OR DEQ, 2006).  Here, we analyze the net effect of anthropogenic stressors by developing statistical models from in-situ data that approximately represent pre-development conditions (pre-1890); post-land and river development conditions (mid 20$^{th}$ century); and post-reservoir management conditions (present-day)."*

3. Line 87-88: The statement is not clear. How warming climate and hotter extremes are linked to land-use changes?

We have rewritten the topic sentence to be more clear and added references from other basins for further reading.  The sentence now reads:

*"Hydrological and land-use changes in temperate-zone river basins are occurring simultaneously with a warming climate marked by hotter extremes (e.g., Cloern et al., 2011, Hamlet & Lettenmaier, 1999, Palmer et al., 2010)"*

4. Line 97-98: What is the characterization of natural variability? How it is linked to climate change?

This sentence was removed during the editing process

5. Line 100-101: What are the natural and background condition? Please mention.

We are not sure what this comment means.  However, these lines were removed during the editing process for clarity.  In other locations, we have strived to be more specific.

6. Line 106: What are chronic and acute anthropogenic factors? Describe with some examples.

These lines were removed during the editing process for clarity.

7. Line 114-120: Remove these results from the introduction section.

Thanks for this suggestion. The introduction has been rewritten, and these results removed.

8. Line 122: Study area will be more appropriate than the setting.

Thanks for this suggestion—the heading to section 2.1 has been changed to "Study Area".

9. Section 2 and its subsequent sections are quite lengthy and not clear. This should be short, precise and reader-friendly. Some results are discussed in this section which should be moved to the result and discussion section.

We agree with the reviewer that this section was too long and complex. We have simplified text by careful editing. Additionally, we reduced section 2 by more than 25% (> 100 lines) by moving some details to the supplement. We have moved Figure 3 and associated text to the beginning of the Results section. Some additional text was also moved to either results or discussion. Note that Figure 2, which was used in the development of the statistical model, has been retained in Section 2.

We also streamlined the results section to focus more clearly on results, both by removing some redundancy and by expanding the discussion section. The discussion now more clearly focusses on the interpretation of results (section 4.1), implications (sections 4.2), and model limitations (section 4.3). Thus, though the discussion section is now larger, it is more clearly organized and major results are more easily found and interpreted.

Importance is given to the derivation of $T_w$ in this paper while the paper title is suggesting the impact of climate change and direct human intervention. Authors can change accordingly.

Our focus on the derivation of Tw was based on our curiosity about how a simple linear regression can model a complex process such as the 1D ADE. The discussion both illuminates why the model is useful but also shows the limitations of the statistical/stochastic modeling approach. We have moved much of this discussion to the supplement, for those that are interested. By moving this portion of the text, we focus more on the results, which do investigate the impact of climate change and direct human interventions. Note we have changed "direct human interventions" to "river system changes" in the manuscript title, to be more clear.

10. Can uncertainty be assessed using RMSE? Any reference to this statement? Or authors can consider separate uncertainty analysis.

We realize that our terminology may have been confusing. We now say (in the last paragraph of section 2.4): *"The skill of each statistical model was assessed by evaluating the root-mean-square error (RMSE) between the composite model estimate and measurements. Our values are compared against the RMSE found between measurements and climatology. "*

Section 3.3.1: How the authors have evaluated the % of $T_w$ change (mentioned in short summary that 30% from system change and 70% from climate change)?

To make this result more clear, we have moved our discussion of % change to section 4.1 and added specific percentages to the sentences. Note that only one significant figure was used in the original short summary.

*"The sum of estimated temperature changes caused by climate, system, and water management changes from ~1900 to the present is ~1.1 ± 0.3 °C (Figure 12) and is consistent with the overall long-term trends in $T_w$ of 1.1 ± 0.2 °C per century (Figure 7a). Of modeled changes since ~1900, 0.81 ± 0.25 °C (74%) is caused by increased $T_a$, while 0.34 ± 0.12 °C (~31%) is caused by alterations in the $T_w$ response to forcing (integrated river system change); river flow alteration produces a -5% change, closing the balance."*

11. Have the authors used any particular separation/attribution analysis? If not then how % of the contribution is shown?

    The attribution analysis was described in section 3.3.1, but has been expanded to be more clear and moved to the Method section (see answers to point 14 and 15 below).

12. Line 746: How the authors used the sensitivity studies? Describe it in methodology.

    To make this more understandable, we have expanded our explanation of how the sensitivity studies are conducted and moved it to the methods section (Section 2.5). The expanded explanation reads:

*"We approximate the influence of changing air temperatures, changing river discharge, and the integrated effect of river system changes through experimentation using our statistical models. The following first-order effects are approximated:*

1. *Climate change impacts: Climate change has driven changes in the 30 year average climatology of daily air temperature in the region (e.g., Mote et al., 2019). We estimate the influence of changed air temperature climatology by running our modern statistical model (model 2000A; see Table 2) using historical downtown climatology (1875-1904) and modern Portland airport climatology (1991-2020) (daily time scale). River flow is kept constant and does not influence results. The difference between these scenarios is attributed to climate change. The uncertainty in modeled $T_w$ is assessed by perturbing input climatology with plausible uncertainty and bias estimates in $T_a$.*
2. *Effect of altered river flow: Changes in river flow seasonality, caused primarily by water resources management but also influenced by changing snow pack (e.g., Naik & Jay, 2011) can influence water temperatures in our 1941 and 2000 era summer models (Table*

*2; river flow was not statistically significant in 1881 era models). The change in the river hydrograph (see Figure 2a) is applied to the 1941 and 2000 era models (Table 2), with the $T_a$ input kept the same between models. The difference in model output shows the influence of altered average river flow on modeled $T_w$ for the July-September time frame between pre-reservoir (1901–1940) and modern (1981–2020) conditions.*

3. *Integrated system changes: Over the past 150 years, multiple landscape and watershed changes, including loss of riparian habitat and reservoir construction, have occurred (Section 2.1). We investigate their net influence on $T_w$ by applying the same river flow and $T_a$ data from 2000– 2020 to models from different eras (Table 2). Because the input into each statistical model is identical, any differences in output $T_w$ are caused by changes in model coefficients (Equation 7). The uncertainty analysis in section 2.4 is applied to determine whether differences are statistically significant, consistent with the hypothesis that river system changes have altered the river's response to external heating and other forcing. "*

13. Line 782: How are the system changes estimated by changing regression coefficients? Please explain.

    The regression coefficients determine the response of the output (water temperature) to inputs (air temperature and river flow). If the same input produces a statistically significant difference in outputs, then the response of the river system to forcing has changed. We explain this now in the methods (section 2.5), as follows:

    *"Over the past 150 years, multiple landscape and watershed changes, including loss of riparian habitat and reservoir construction, have occurred (Section 2.1). We investigate their net influence on $T_w$ by applying the same river flow and $T_a$ data from 2000– 2020 to models from different eras (Table 2). Because the input into each statistical model is identical, any differences in output $T_w$ are caused by changes in model coefficients (Equation 7). The uncertainty analysis in section 2.4 is applied to determine whether differences are statistically significant, consistent with the hypothesis that river system changes have altered the river's response to external heating and other forcing"*

14. In Section 2, several anthropogenic factors are discussed. The authors should consider these factors in attribution analysis.

    As we state in the last paragraph of section 3.3 and elsewhere, we are not able to discern individual anthropogenic factors (e.g., water diversions vs. land use change vs. shading) using our approach. We have added the following sentence to section 4.3 to reiterate this point: *"A numerical modeling approach is needed to isolate individual anthropogenic stressors and to determine how landscape and climate changes can influence $T_w$ in incremental, nonlinear, and interdependent ways (e.g., Berger et al., 2004)."*

15. What is the significance of precipitation in this work? As precipitation is an important climatic variable, it can't be ignored in this analysis. The authors can refer following articles. (* Swain, S. S., Mishra, A., Chatterjee, C., & Sahoo, B. (2021). Climate-changed versus land-use altered streamflow: A relative contribution assessment using three

complementary approaches at a decadal time-spell. Journal of Hydrology, 596, 126064. * Liang, S., Wang, W., Zhang, D., Li, Y., & Wang, G. (2020). Quantifying the impacts of climate change and human activities on runoff variation: case study of the upstream of Minjiang River, China. Journal of Hydrologic Engineering, 25(9), 05020025.)

It is beyond the scope of this study to investigate precipitation or the reasons for altered river discharge. For interested readers, we included some additional references such as Hamlet et al. (1999), Cloern et al. (2011), and the two suggested above. The reviewer is correct that both climate change and land-use cause changes to run-off, and we did mention this in the original submission. We trust that the additional references will highlight the issue more clearly. The two suggested references are found in the second-to-last paragraph in section 3.

We also clarify that the effects of precipitation are included in the river discharge term and are therefore included in our statistical model. As already reviewed in section 2, the river discharge is caused both by direct runoff, especially during winter, and primarily by snowmelt and groundwater in the summer. Including precipitation as a variable in our regression would be redundant. We now state in the model development section (second paragraph, section 2.3):

*"The river discharge term incorporates the net influence of precipitation, snowmelt, and groundwater recharge."*

16. Figures should not be cited in the conclusion. Please rewrite this section.

The references to figures have been removed.

17. Line 902: The citation of Jay and Naik, 2011 is wrong. Citations should be in a uniform manner followed by HESS guidelines.

We did some light editing on the punctuation for this reference. We also checked all the other references and made additional punctuation updates where needed. We also added some DOI information that was missing to some references. To the best of our ability, the references now follow the format the HESS guidelines (https://www.hydrology-and-earth-system-sciences.net/submission.html).

18. Proper proof-reading is needed, but more importantly, better use of technical language and precise description is lacking throughout the manuscript.

We have gone through the manuscript and made it more succinct, where possible. During the revision we removed redundant information and clarified many passages. We made more specific many sentences and removed some words that were unclear or that could be interpreted as non-technical. We believe we have used appropriate technical language and description throughout. However, we have also aimed for overall clarity and have tried to avoid the use of jargon, following best practice for technical writing. We note that the other reviewer did not comment on any writing style issues, and only found a couple of typos. These have been fixed. Neither we nor the other reviewer find any systematic or widespread problem with language or

proofreading. Without a precise description or specific examples, we have done all we can to address this comment.

---

## Author Comment (AC2)

Dear Editors and Reviewers,

We have closely considered the comments of both reviewers and have undertaken a major rewrite of the text to improve our presentation and make the scientific novelty more clear. We thank both reviewers for their comments, which have greatly improved the manuscript. Pending approval from the editors, we will upload the updated manuscript. In the mean-time, we have outlined our edits and include details about specific improvements.

As described in the detailed responses below, the introduction is now more succinct and focusses more clearly on broad relevance and scientific questions. Much of the locally specific material has been moved to either section 2.1 (Methods>Study Area) or to a supplement. The methods section has been reduced by more than 25%, both by careful editing but also by moving supporting information to a supplement. The results section has been streamlined to more clearly focus on a description of results, following scientific convention. The discussion has been reorganized and interpretation and analysis has been moved there from other sections. The reorganization more clearly focuses on interpretation (section 4.1), implications (section 4.2), and limitations (section 4.3). Where requested, additional explanations and specificity have been added. Overall, the text is more than 100 lines shorter.

Through the careful rewrite, the modeling approach and novel contributions are more clearly findable and understandable, and the methods are more succinctly and clearly explained. The concerns of both reviewers concerns about the length and understandability of the manuscript have therefore been addressed. Additional comments and details are given in the response to the first reviewer.

Best regards, also on behalf of co-authors,

Stefan Talke

**Response to Reviewer 2**

We thank the reviewer for their positive and constructive comments; they have improved the text. Replies to comments are shown in blue below.

**General Comments:**

The manuscript gives a thorough investigation of changes in stream temperature in the Willamette river from the 1850s to the present. The authors compile air temperature, discharge, and stream temperature data from a variety of sources throughout the region, and use these data to construct statistical models that give insight into the magnitude of change in stream temperature throughout three different historical periods. The authors then investigate the seasonal and interannual changes in observed stream temperature records and validate their models by comparing model accuracy (including a comparison to other stream temperature

predictions in the same region.) The models are then used to quantify the importance of climatic and system changes (notably reservoirs, loss of shading, and landscape alterations) over time, with good quantification of how the two compare in magnitude and seasonality using sensitivity experiments.

The results of this paper give a clear insight into how and why stream temperatures are changing in the Willamette river, investigate the causes of these changes, and draw connections to the ecological impacts of rising stream temperatures.

Thanks for the positive comments.

While the paper is well framed, the extensive methods section makes it difficult for the reader to keep track of the different data sources and to get a clear and concise understanding of how the models presented in the results were constructed. Thus, the manuscript could benefit from making the methods section more concise, contextualizing the model with other models that have been used for stream temperature modeling, with additional information made available in supplemental materials.

We agree that the Methods section was too long. As also described in the response to the other reviewer, we have made the suggested changes. We have reduced the introduction and methods section by more than 130 lines, moving some material to a supplement and eliminating it altogether. We have also edited the text for clarity. The remaining text reads more succinctly and clearly, in our opinion. As suggested below, we have added more references and included a discussion of the typical RSME found in other statistical and data-driven models, to help contextualize our results. We note that many temperature studies were already referenced in the original submission, including Caissie et al., 1998; Benyaha et al., 2007, Scott (2020), Moore (1967), Donato (2002), Bottom et al. (2011), Mayer (2012) and many others. The more compact methods section helps highlight these references better.

Overall, the model results and analysis substantiate the manuscript's conclusions about changing temperatures in the Willamette river over time. Finally, the digitization of historical observations since the mid-19th century adds value as data are sparsely available for these time periods, but are not yet archived for public use, detracting from the overall impact of the paper.

We agree that the archival measurements are valuable, and one of the potential broad impacts of this research effort. They are potentially valuable for future process-based modeling efforts, and for comparisons with other systems. Our intention has always been to make the measurements available, pending the review process. They are now available: https://doi.org/10.15760/cee-data.06

**Specific Comments:**

1. The authors provide extensive detail about the specific climate and long-term changes in the region. Because the detail of different regions is so extensive, it is difficult for a reader who is not familiar with the study area to distill the historical changes of the region to understand how system impacts have changed over time.

This is a good point.  We aimed to be thorough and to show that much is already known about land-use changes, but may have inadvertently obscured the big picture.  To understand why water temperatures are shifting, we still think it is important to show how, why, and when shifts occurred to the riparian corridor, the geometry of the river, and water surface area.  But we have now moved most of these details to the "Methods>Study Area" area section or to a supplement, and focus the introduction more on the 'big picture' and scientific relevance (see below and the comments to the other reviewer).  This way, readers who are interested in the detail can find it, but it is not essential to the motivation and 'flow' of the paper.

> A more concise description would help the reader to focus on key points which are integrated into the model methodology. Additionally, contextualizing the changes in the Willamette river with other river basins in the continental U.S. (e.g. other basins with similar snowmelt influence) in the introduction (rather than just referring to similar studies in the discussion) could help a broader audience understand the study area before getting into methods and results.

We have simplified and reduced the size of the introduction and have rewritten several paragraphs to more clearly frame that the major changes to flow regime, air temperature, and land use that have occurred in the Willamette have also occurred elsewhere. As described above, we have also made the Methods section more concise.

1. Section 2.3: While the discussion of the advection-diffusion equation provides important information about how physical understanding can help inform accurate statistical model architecture, the in-depth analysis detracts from the reader's ability to understand the equations used for the final statistical models that are run and used for reporting results. Shortening sections 2.3 and 2.4 and/or highlighting what equations relate to the final chosen models will greatly help the readability of the methods

We have greatly simplified section 2.3 and 2.4 and moved some of the material to a supplement. We believe the shortened section is much more readable and understandable.

1. Lines 509-510: The authors refer to the "total of 8 statistical models" that are developed (listed in Table 2). When the reader refers to Table 2, however, there are 7 different stations named, implying that there are not 8 but 7 different models. Furthermore, it is difficult to decipher the governing equations for these models, and if the only major difference is the data sources and time periods used for each. Clarification would help greatly in this section of the manuscript.

Thanks for pointing out this potential source of confusion and for catching the typo about the number of models (originally there were 8, but we removed a model based on 2000-2015 Vancouver measurements because the station was moved after 1966, leading to bias). We have also clarified which equation we use for all the models. Indeed all models use the same basis function and they diffes primarily in data source and calibration time period. We also now explain our naming convention, which should emphasize that the main difference is the data and time period used. The new description reads:

*"A total of 7 statistical models are developed from Equation 7, using data from the 19th century (1881–1890), mid-20th century (1941–1952), and modern period (2000–2015) (see Table 2). The models differ in the location of air temperature data and time period used. These three calibration periods were chosen based on available data; they approximate (nearly) pre-development conditions, pre-flood control conditions, and modern conditions. The models are named based on the first year of calibration data and the first letter of the meteorological station used; for example, 1941V and 1941D are models trained with 1941–1952 data from Vancouver and Downtown Portland, respectively (Table 2).* "

1. While there is extensive discussion of how the chosen statistical model was derived, there is no noted comparison to other statistical stream temperature methods making it difficult to put this model in the context of previous statistical models which are also derived from physics-based equations.

Thanks for noting this oversight. We have added the following sentence to the third paragraph of results section 4.1:

*"Our results compare well with traditional linear regression and stochastic models, which have reported RMSE of ~0.6–1.9°C, depending on model type, river size and location, and averaging period (e.g., Caissie 1998; see also review by Benyahya et al., 2007 and references therein). More recent statistical models, including air2stream (Toffolon and Piccolroaz, 2015) and machine learning approaches (e.g., Fiegl et al., 2021), report RMSE of 0.5-1 °C on a daily scale, similar to the results presented here (Table 2). Results are also comparable to numerical models that generally have an RMSE <1°C (e.g., Dugdale et al., 2017).*

Adding information on how this model compares to other statistical models (such as air2stream, ARIMA models etc.) would help give a better understanding and context to readers familiar with stream temperature modeling, and help make these methods more applicable to other systems.

We agree that framing our model skill with respect to other models and model types is important, and have added the sentences above. However, a detailed comparison to other model types and approaches is beyond the scope of this effort, and we have therefore kept our comparison general, since more specificity would also require introducing/explaining those models.

1. The authors mention 8 different models (line 509), however, the text does not clearly explain what differentiates these 8 different models. iTable 2 includes 7 different models. Including a table of the different models, and some measure of comparative accuracy for the summer vs. winter sub-models would be helpful.

Thanks again for catching the typo about the 8 different models. This has been changed, and we have expanded our explanation of how the models differ (see above). Note that we already

included both daily and monthly-averaged RMSE in Table 2 for the summer, winter, and annual sub-models, so there is already some measure of comparative accuracy. However, we have rearranged the table so that the model name is on the left hand side of the table. In this way, it is now more obvious that each row refers to a model that is calibrated to a different location and/or set of years.

1. Line 969: The manuscript states that "The water temperature data used in the research is available upon request, and will be uploaded to a data repository upon acceptance of the manuscript." With other meteorological and flow data available from other sources. Because the procurement of data used in this study is a large value add, publishing all data in a corresponding data package together (if possible) will greatly improve the open use of data from this study.

Our intention has always been to make the measurements available, pending the review process. These measurements are now available: https://doi.org/10.15760/cee-data.06.

**Technical Corrections:**

Line 245: Missing space– should read "1881– 2021 record"

This has been fixed.

Table 2: 10th column (RMSE Winter Calibration) Does not properly align/format with 11th column should be fixed.

Thanks for catching! This has been fixed.

Figure 3: the x-axis of the bottom right-hand figure (d) is cut off and should be fixed.

Fixed.

**Citation**: https://doi.org/10.5194/egusphere-2022-793-RC2

---

## Referee Report (RR1)

The Manuscript *Warming of the Willamette River: 1850–present: The effects of climate change and river system alterations* has undergone significant changes which have largely improved the readability and therefore impact of the manuscript. The introduction now gives the reader an understanding of why the data and models presented in the manuscript augment existing stream temperature modeling studies. It also provides clear context as to why the new pre-development data, which are not available in many other locations, can give valuable insight into how the river has changed with clear linkage to how these insights may be applicable to other study areas with similar properties.

The updated manuscript greatly simplifies the methods allowing the reader to better understand a) the description of the chosen statistical model itself b) the different data sources used to train 7 different models presented in the manuscript, and c)  how these models are used for the interpretation of results. The results and conclusions sections, now more concise, give the reader a clear idea of the study's implications about how temperate streams respond to climate and stream alterations over time.

Finally, with archived observational and modeled data now publicly available under the listed DOI, the study presents a comprehensive package for understanding and studying long-term stream temperatures in the region. I now believe that this manuscript would provide a valuable contribution to the journal's readers.